# Could teacher-perceived parental interest be an important factor in understanding how education relates to later physiological health? A life course approach

Camille Joannès[1]*, Raphaële Castagné[1], Benoit Lepage[1,2], Cyrille Delpierre[1], Michelle Kelly-Irving[1,3]

**1** Equity Research Team, CERPOP, Université de Toulouse, Inserm, Université Paul Sabatier, Toulouse, France, **2** Department of Epidemiology, CHU Toulouse, Toulouse, France, **3** Interdisciplinary Federal Research Institute on Health & Society (IFERISS), Université Toulouse III Paul Sabatier, Toulouse, France

* camille.joannes@univ-tlse3.fr

**Data Availability Statement:** Regarding data ACCESS, NCDS data are available by registering on the UK data service repository https://

## Abstract

Education is associated with later health, and notably with an indicator of physiological health measuring the cost of adapting to stressful conditions, named allostatic load. Education is itself the result of a number of upstream variables. We examined the origins of educational attainment through the lens of interactions between families and school i.e. parents' interest in their child's education as perceived by teachers. This study aims to examine whether parental interest during a child's educational trajectory is associated with subsequent allostatic load, and whether education or other pathways mediate this relationship. We used data from 9 377 women and men born in 1958 in Great Britain and included in the National Child Development Study to conduct secondary data analyses. Parental interest was measured from questionnaire responses by teachers collected at age 7, 11 and 16. Allostatic load was defined using 14 biomarkers assayed in blood from a biosample collected at 44 years of age. Linear regression analyses were carried out on a sample of 8 113 participants with complete data for allostatic load, missing data were imputed. Participants whose parents were considered to be uninterested in their education by their teacher had a higher allostatic load on average in midlife in both men (β = 0,41 [0,29; 0,54]) and women (β = 0,69 [0,54; 0,83]). We examined the role of the educational and other pathways including psychosocial, material/financial, and behavioral variables, as potential mediators in the relationship between parental interest and allostatic load. The direct link between parental interest and allostatic load was completely mediated in men, but only partially mediated in women. This work provides evidence that parents' interest in their child's education as perceived by teachers is associated with subsequent physiological health in mid-life and may highlight a form of cultural dissonance between family and educational spheres.

ukdataservice.ac.uk/ (Persistent identifier (DOI): 10.5255/UKDASN-5560-4 and 10.5255/UKDA-SN-5594-2). However, the biomedical survey used for this research contains sensitive data and therefore added restrictions because of its sensitive nature hence requesting a special license. Qualified readers can access the biomedical survey data via the UK data service with a special license, in the same way that we obtained the data, but they are not allowed to share it.

**Funding:** MKI received funding from the European Research Council (ERC) under the European Union's Horizon 2020 research and innovation programme (grant agreement No. [856478]). CJ received funding from the Institut National du Cancer & the Institut de recherche en santé publique is (grant No. INCA-CA-2019-204).

**Competing interests:** The authors have declared that no competing interests exist.

# 1. Introduction

One of the most consistent findings in the field of social epidemiology is that educational attainment is associated with health. Across countries, and over time, lower educational attainment has been associated with poorer health outcomes [1]. These associations are often explained by the fact that better educated people are less likely to experience the disadvantaged material conditions or psychosocial distress caused by economic hardship and tend to have healthier lifestyles compared to the less educated [2]. However, Gallo et al. showed that health behaviors and lifestyle factors explained only a part of the educational inequalities in total mortality of members of European cohorts [3]. Beyond these groups of mechanisms others have been examined, such as environmental exposures and material conditions [4, 5], calling for further studies to examine the mechanisms through which education relates to health.

The concept of embodiment rests upon a key set of mechanisms likely to underlie the relationship between education and health. This concept "refers to how we, like any living organism, literally incorporate, biologically, the world in which we live, including our societal and ecological circumstances" [6]. Embodiment can occur through several mechanisms including the internal physiological response to social exposures, among which feature intra-familial relationships and social interactions. Such interactions are perceived through the senses, interpreted by the central nervous system leading to peripheral physiological responses [7]. These physiological responses are adaptive processes which maintain physiological stability in response to environmental challenges [8]. The repeated activation of compensatory physiological mechanisms as a result of chronic exposure to stress can lead to physiological wear-and-tear, termed allostatic load. Allostatic load measures the consequence of a prolonged activation of the stress response system by external challenges, leading to physiological imbalances across systems [9]. Previous research has shown that allostatic load is associated with physical and functional decline, cardiovascular events, and mortality [10, 11]. A few studies have examined the association between education and allostatic load [12, 13] indicating that a higher level of education is associated with a lower allostatic load, only partly explained by different potential mediators (including health, behavioral and psychosocial factors) suggesting that an important part of this association remains to be explained.

Life course research indicates that the dynamic processes of adaptive allostasis most likely begin in early life [14, 15]. One recent study documented that disadvantaged early life socioeconomic conditions are associated with an increased risk of having a higher allostatic load in midlife, mainly through educational pathways [16]. Elsewhere, Hamdi et al. reported that the relationship between education and allostatic load may be partly explained by family influences [17]. As such, educational attainment "is an excellent marker of the 'healthfulness' of accumulated childhood experience" [18] as the social environment in early life may therefore have lasting effects on different social, biological, and behavioral factors that might act as mechanisms connecting education to repeated stress, which in turn affects health [19]. Because educational attainment is the end point of a complex process where a wide array of factors (institutional, interpersonal, individual) shape trajectories of schooling, there is a need to look upstream in order to capture education as a long-term process grounded in a broader social context [19]. However, it remains as yet unclear which elements of the early life environment upstream of education are likely to be involved in the embodiment dynamic, leading to physiological wear-and-tear.

An analysis of the 1958 British Birth Cohort Study suggested that parental interest in their child's education could be an early life factor contributing to self-rated health at 33 years of age [20]. More broadly parental interest in their child's education has also been reported to have positive effects on psychosocial adjustment [21] later mental health [22] and metabolic

outcomes in middle-age [23]. Moreover, parental interest in their offspring's schooling has been identified as a determinant of educational success [24–27]. This parental involvement in children's academic socialisation, has been shown to influence academic success over and above child's social class and cognitive abilities [28]. Consequently, parental interest in their child's education could be a variable to consider, when examining how the relationship between the home and school environments in early life affects physiological wear-and-tear, through educational attainment.

Parental interest in their child's education can be reported by the parents themselves, reflecting their perception of their own involvement in the upbringing of their children. It can also be reported by teachers. Teacher's assessments provide one perspective of the situation which may partly reflect the position and viewpoint of the educational institution in terms of children's compliance with academic requirements and potentially capture the tension between the home and school environments experienced by some children [29]. This dissonance that children may experience when exposed to different family and school environments, may lead to challenges when adapting to school. Such an early life stressor may lead to the chronic solicitation of children's physiological stress response systems, in turn affecting subsequent physiological health.

A previous study in a Swedish cohort examined the association between parental academic involvement as perceived by teachers and allostatic load in midlife. Rather than parental social class or availability of practical academic support, parental interest in their children's studies during the last year of school was found to predict adult allostatic load, largely mediated by academic achievement, which the authors attributed to the potential consequence of physiological stress over the life course [30]. Our study aims to examine the association between parents' interest in their child's education as perceived by teachers, and multi-system physiological dysregulation as measured by allostatic load, and whether education and other pathways mediate this relationship.

We hypothesize that parents' interest in their child's education as perceived by teachers is an indicator of academic socialisation that may capture early life stressors and thus be related to later physiological wear-and-tear, partly through education and other pathways. In this study, we take a life course approach to (i) test whether parental interest is associated with allostatic load, and (ii) explore the educational and other pathways through which parental interest may be differentially embodied during childhood, adolescence and early adulthood, leading to physiological wear-and-tear, as measured by allostatic load.

## 2. Materials and methods

### 2.1. Study population

Our study is based on secondary analyses of data from the 1958 National Child Development Study (NCDS), an observational prospective population cohort study, which included all live births in Great Britain during one week in 1958 (n = 18 555). The NCDS has been described in detail elsewhere [31]. Data collection on health, economic, social and developmental factors was carried out on cohort members from birth until now at age 7, 11, 16, 23, 33, 42, 44/45, 46, 50, 55 and 62 years and conducted by the Centre for Longitudinal Studies. Written informed consent was obtained from parents for childhood measurements and ethical approval for the adult data collection was obtained from the National Research Ethics Advisory Panel. When cohort participants were 44 and 45 years of age, a biomedical survey was conducted including a self-reported questionnaire, blood and saliva samples as well as anthropometric measurements with data available for 9 377 individuals. Ethical approval for the age 45 survey was given by the South East Multicentre Research Ethics Committee. Participants in this survey were found to be representative of the general cohort [32]. A total of 1 264 participants were excluded from our

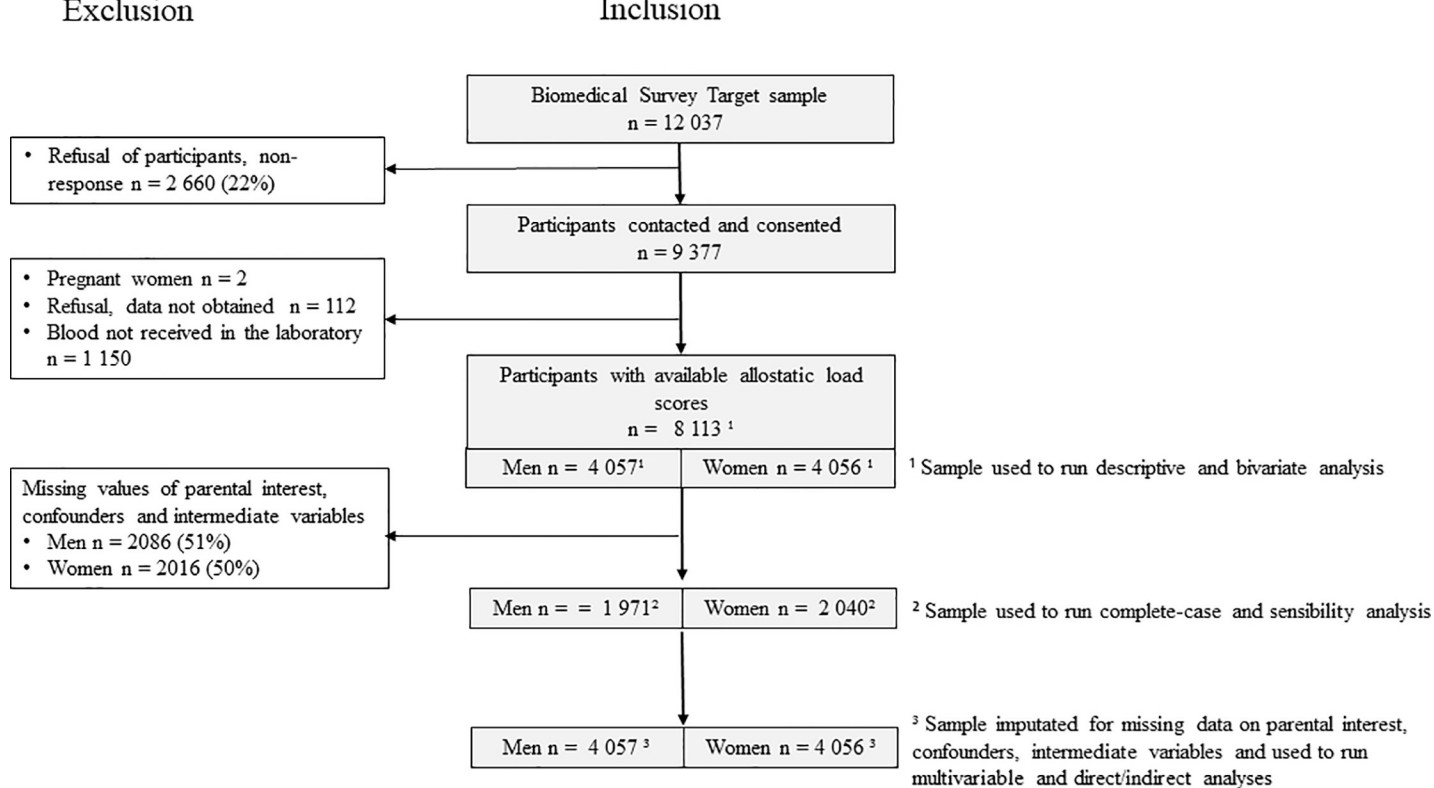

**Fig 1. Diagram of inclusion and exclusion criteria for the analysis from the biomedical survey of the NCDS 58.**

analyses, including pregnant women and those from whom blood was not obtained, leaving 8 113 participants. Our sample selection flow-chart is presented in Fig 1. NCDS data are open access datasets available to non-profit research organizations through the UK Data Service.

## 2.2. Allostatic load at age 44–45

The allostatic load score was constructed based on previous work using the NCDS the same initial definition of allostatic load [33]: in order to represent four physiological systems, 14 available biomarkers were used: the neuroendocrine system (salivary cortisol t1, salivary cortisol t1–t2); the immune and inflammatory system (insulin-like growth factor-1 (IGF1), C-reactive protein (CRP), fibrinogen, Immunoglobulin E (IgE)); the metabolic system (high-density lipoprotein (HDL), low-density lipoprotein (LDL), triglycerides, glycosylated hemoglobin (HbA1C)); the cardiovascular and respiratory systems: (systolic blood pressure (SBP), diastolic blood pressure (DBP), heart rate, peak expiratory flow). Using sex-specific quartiles, each biomarker was dichotomized into "high" (coded as 1) and "low" (coded as 0) risk. The sum of these 14 dichotomized biomarkers resulted in an overall allostatic load score ranging from 0 to 14, where a higher score represented a higher allostatic load, and a greater health risk. We also recoded allostatic load score into a 3 category variable where a score of 0–2 was considered to be "low", 3–4 as "middle", and 5–14 as "high" based on tertiles in the sample [34].

## 2.3. Parents' interest in their child's education as perceived by teachers

Parental interest was measured at ages 7, 11 and 16 using information provided by the child's teachers. The teacher was asked to report the level of interest of each parent in their child's

education through closed questions with four possible answers: Overly concerned; Very interested; Some interest; and Little interest. Based on this, we created a new binary variable for parental interest aiming to identify parents "interested" or with"low/no interest" in their child's education, according to the teacher. We grouped the "overly concerned" and "very interested" categories together to represent the "interested" category, while grouping the "some interest" with "little interest" categories together to represent "low/no interest". We hypothesized that interest from both parents at any one age belongs to the category "interested". However, if only one of the parents was considered to be interested or if neither were, we considered this to belong to the category "low/no interest". We conducted a sensitivity analysis to examine the stability of parental interest, using a series of regression analyses to identify whether changes to the categories (Overly concerned; Very interested; Some interest; and Little interest) had an effect on the association with allostatic load. We observed no change to the results (S1 Table).

## 2.4. Prior confounders

To examine the relationship between parental interest and allostatic load, prior confounding variables potentially associated with both parental interest and allostatic load were added to the multivariable models. We selected variables most likely to be social or biological confounding factors from questionnaires completed during childhood by parents of cohort members, based on the literature.

Socioeconomic confounders were included using parental socioeconomic classification of occupations (SEC), via a questionnaire completed at birth (I-professional occupations & II-intermediate occupations/III-skilled occupations (non-manual)/III-partly skilled occupations (manual)/IV-partly skilled occupations & V-unskilled occupations) and using information on material living conditions, collected at ages 7, 11 and 16 (advantaged/disadvantaged).

Confounding factors related to parental education were also included using parental educational attainment at birth for the mother and at 7 year for the father (left school <14 year/ lefts school ≥15 year), and parenting practices including reading to the child and outdoor activities, measured at age 7 ("Frequent/Occasionally, Hardly ever) [35].

Other prior confounding variables were also selected. At ages 7, 11, and 16, a binary adverse childhood experiences variable (ACEs) was constructed, as well as a binary childhood pathologies variable. Using data collected at age 7, a birth order variable was created (Single child/ Eldest/2nd place or more), and an assessment of the child's cognitive ability at age 7 (Copy-a-Design test where scores range between 0 and 12). See S1 File for more information about early life confounder variables.

## 2.5. Intermediate life course variables

In order to determine whether any observed associations between parental interest and allostatic load were due to subsequent adult intermediate factors, we selected four intermediate groups of life course variables as pathways between parental interest and health, based on epidemiological evidence and on empirical studies. The following mediating factors were then added to the models: i) Parental interest could promote a positive accumulation of academic and social success [36], reducing the impact of economic difficulties on educational attainment [37]. We therefore added the respondent's: educational attainment at age 23 (A level-12 years of education/O levels-10 years of education/ No qualifications), occupational social class at 33 y (I-professional occupations & II-intermediate occupations/III-skilled occupations (non-manual)/III-partly skilled occupations (manual)/IV-partly skilled occupations & V-unskilled occupations) and wealth at age 33 using a wealth variable based on information about home

ownership and the price of the house adjusted for economic inflation of the year of purchase (owner highest price/owner high price/owner median price/owner lowest price/not owner]; ii) Tension between home and school environments experienced by some children may provide an insecure social environment, contributing to health through psychological processes whereby resulting negative emotions through psycho-neuroendocrine mechanisms could lead to a worse health status [38]. To capture this, we added psychological/psychosocial status [malaise inventory at age 23 (No psychological distress/ psychological distress); sense of personal control at age 33 (Internal/external)]; iii) The adoption of health behaviors is closely related to education, whereby people with a higher education level are more likely to have protective health behaviors, impacting physiological functioning [16, 39]. Health behaviors at age 42 were considered as a proxy for behavioral patterns in adulthood included self-reported physical activity, alcohol consumption and smoking status.

We defined the sets of mediators according to the temporal and causal assumptions of the life course framework. As such, we made a pragmatic methodological assumption about the temporal ordering of variables in order to examine the mediating pathways whereby the exposure variable (parental interest at age 7, 11 and 16) preceded the respondent's education, which preceded the other intermediate variables (psychological distress, socioeconomic position and health behaviors).

## 2.6. Statistical analysis

Our analyses were stratified by sex since behavior at school differs between girls and boys. Girls tend to be more compliant with institutional rules, facilitating the teacher's task, boys are more frequently in conflict between academic expectations and their socially recognized particularities. Also, this allows us to consider sex/gender differences in health.

First, descriptive and bivariate statistics were carried out, using the chi-squared test for categorical variables and Wilcoxon signed-rank test or Student t-test for continuous variables. We considered allostatic load as a categorical variable in three groups, in order to ascertain any association between the covariates and allostatic load. Second, to study the association between parental interest and allostatic load, regression coefficients and 95% confidence intervals (CI) were estimated using linear regressions where allostatic load was entered as a continuous variable. We compared regression coefficients across nested models to observe the change in effect according to subsequent adjustments.

i. **To study the link between parental interest and allostatic load**

- Model 1: Linear regression between parental interest and allostatic load

- Model 2: Model 1 plus baseline confounders (parental SEC, material living conditions, level of parental education, reading and outdoor activities, ACEs, health problems in childhood, sibling order, cognitive ability).

ii. **To qualify pathways mediating the relationship between parental interest and allostatic load**
In these models, we considered that the last set of mediators had an important mediating role if the change of the regression coefficient characterizing the association between parental interest and allostatic load was large and if these mediators were associated to allostatic load.

- Model 3: Model 2 plus educational level

- Model 4: Model 3 plus psychosocial/psychological variables (Sense of personal control, malaise)

- Model 5: Model 4 plus socioeconomic status/financial variables (Occupational social class and wealth)

- Model 6: Model 5 plus health behaviors variables (smoking, alcohol consumption, physical activity)

Third and finally, to disentangle and quantify the direct and indirect effect for parental interest on allostatic load, we carried-out different linear regression steps in both men and women, to analyze mediation, applying the method of VanderWeele and Stijn Vansteelandt [40]. See S2 File for more information on the different steps of the analysis of estimated direct and indirect effects.

In order to control for potential biases due to missing data, multivariable analyses were conducted on the multiply imputed data using the ICE method available in Stata ®v14. Twenty imputations were performed assuming that the data were missing at random (MAR). Comparisons were then made between complete-case multivariable analyses and multivariable analyses based on imputation estimates (S2 Table).

## 3. Results

Descriptive statistics for men and women on the complete-case data are provided in S3 Table. The majority of our sample (78% in men and 75% in women) had a low [0–2] or medium [3–4] allostatic load in midlife. Additionally, during childhood, 47% of the cohort members' parents were perceived by the teacher as taking interest in their child's education, while 42% were described as uninterested or not very interested.

The bivariate analyses by allostatic load group on the complete-case data are provided for men in Table 1A and women in Table 1B. In both sexes, we observed a higher proportion of cohort members whose parents were deemed uninterested in their child's education by the teacher in the high allostatic load group, compared to the medium and low allostatic load group. Regarding the early life social environment, both men and women with a high allostatic load at age 44 were more likely to have had parents with low SEC, lived in disadvantaged material conditions, had a father or a mother who left school before the age of 14, been exposed to adversity and to have had a lower score on the copy-a-design test, compared to those with medium and low allostatic load. In addition, men with a high allostatic load at age 44 were more likely to have been an only child, to have engaged occasionally or rarely in outdoor activities and to have had health problems, than those with medium and low allostatic load. Regarding intermediate variables in adulthood, both men and women with a high allostatic load at 44y were more likely to have had a low level of education, an external sense of personal control, a lower social position (occupational social class and wealth), to smoke heavily, to be heavy drinkers or abstainers and to have had a low level of physical activity, than those with medium and low allostatic load. In addition, women with a high allostatic load at 44y were more likely to report a psychological distress, than those with medium and low allostatic load.

Bivariate analyses by parental interest are reported in Table 2. Parental interest was associated with all childhood variables, and all intermediate variables. In relation to our hypothesis on the educational and others pathways, women and men who had a low level of education, psychological distress at age 23, an external sense of personal control at age 33, a low social position at age 33 (occupational class and wealth), and who were smokers, heavy alcohol drinkers or abstainers and had low physical activity, were more likely to have had parents who were described as uninterested or not very interested in their children's education by the teacher.

**Table 1. Bivariate statistics on the complete-case sample (n = 8 113): Allostatic load categories according to confounding and intermediate variables in (A) men and in (B) women.**

| | Allostatic load at 44y | | | | | | | |
|---|---|---|---|---|---|---|---|---|
| | A. Men n(%) n = 4 075 (50%) | | | | B. Women n(%) n = 4 056 (50%) | | | |
| | Low n(%) | Medium n(%) | High n(%) | P-value* | Low n(%) | Medium n(%) | High n(%) | P-value* |
| | 1793 (44.20%) | 1386 (34.16%) | 878 (21.64%) | | 1824 (44.97%) | 1235 (30.45%) | 997 (24.58%) | |
| **Parental interest (7-16y)** | | | | | | | | |
| Both interested | 930 (51.87%) | 634 (45.74%) | 348 (39.64%) | <0.001 | 989 (54.22%) | 555 (44.94%) | 362 (36.31%) | <0.001 |
| Low/No interest | 698 (38.93%) | 578 (41.70%) | 427 (48.63%) | | 654 (35.86%) | 524 (42.43%) | 488 (48.95%) | |
| Missing | 165 (9.20%) | 174 (12.55%) | 103 (11.73%) | | 181 (9.92%) | 156 (12.63%) | 147 (14.74%) | |
| *Prior confounders* | | | | | | | | |
| **Parental SEC (birth)** | | | | | | | | |
| I & II | 405 (22.59%) | 240 (17.32%) | 103 (11.73%) | <0.001 | 414 (22.70%) | 184 (14.90%) | 108 (10.83%) | <0.001 |
| IIINM | 196 (10.93%) | 117 (8.44%) | 67 (7.63%) | | 201 (11.02%) | 110 (8.91%) | 80 (8.02%) | |
| IIIM | 805 (44.90%) | 666 (48.05%) | 442 (50.34%) | | 789 (43.26%) | 613 (49.64%) | 509 (51.05%) | |
| IV&V | 295 (16.45%) | 262 (18.90%) | 211 (24.03%) | | 309 (16.94%) | 248 (20.08%) | 240 (24.07%) | |
| Missing | 92 (5.13%) | 101 (7.29%) | 55 (6.26%) | | 111 (6.09%) | 80 (6.48%) | 60 (6.02%) | |
| **Material living conditions (7y)** | | | | | | | | |
| Advantaged | 1191 (66.42%) | 804 (58.01%) | 500 (56.95%) | <0.001 | 1158 (63.49%) | 735 (59.51%) | 570 (57.17%) | <0.001 |
| Disadvantaged | 337 (18.80%) | 360 (25.97%) | 241 (27.45%) | | 399 (21.88%) | 308 (24.94%) | 300 (30.09%) | |
| Missing | 265 (14.78%) | 222 (16.02%) | 137 (15.60%) | | 267 (14.64%) | 192 (15.55%) | 127 (12.74%) | |
| **Father's level of education (7y)** | | | | | | | | |
| Left school ≥15y | 451 (25.15%) | 287 (20.71%) | 135 (15.38%) | <0.001 | 487 (26.70%) | 252 (20.40%) | 167 (16.75%) | <0.001 |
| Left school <14y | 1101 (61.41%) | 875 (63.13%) | 598 (68.11%) | | 1067 (58.50%) | 795 (64.37%) | 677 (67.90%) | |
| Missing | 241 (13.44%) | 224 (16.16%) | 145 (16.51%) | | 270 (14.80%) | 188 (15.22%) | 153 (15.35%) | |
| **Mother's level of education (birth)** | | | | | | | | |
| Left school ≥15y | 516 (28.78%) | 325 (23.45%) | 182 (20.73%) | <0.001 | 552 (30.26%) | 300 (24.29%) | 173 (17.35%) | <0.001 |
| Left school <14y | 1189 (66.31%) | 971 (70.06%) | 640 (72.89%) | | 1172 (64.25%) | 858 (69.47%) | 766 (76.83%) | |
| Missing | 88 (4.91%) | 90 (6.49%) | 56 (6.38%) | | 100 (5.48%) | 77 (6.23%) | 58 (5.82%) | |
| **Reading activities (7y)** | | | | | | | | |
| Every week | 920 (51.31%) | 687 (49.57%) | 423 (48.18%) | 0.263 | 893 (48.96%) | 590 (47.77%) | 483 (48.45%) | 0.832 |
| Occasionally | 488 (27.22%) | 355 (25.61%) | 240 (27.33%) | | 548 (30.04%) | 389 (31.50%) | 304 (30.49%) | |
| Hardly ever | 173 (9.65%) | 142 (10.25%) | 94 (10.71%) | | 168 (9.21%) | 98 (7.94%) | 90 (9.03%) | |
| Missing | 212 (11.82%) | 202 (14.57%) | 121 (13.78%) | | 215 (11.79%) | 158 (12.79%) | 120 (12.04%) | |
| **Outdoor activities (7y)** | | | | | | | | |
| Most weeks | 1402 (78.19%) | 1017 (73.38%) | 657 (74.83%) | 0.032 | 1449 (79.44%) | 951 (77.00%) | 772 (77.43%) | 0.41 |
| Occasionally/hardly ever | 182 (10.15%) | 169 (12.19%) | 101 (11.50%) | | 161 (8.83%) | 130 (10.53%) | 105 (10.53%) | |
| Missing | 209 (11.66%) | 200 (14.43%) | 120 (13.67%) | | 214 (11.73%) | 154 (12.47%) | 120 (12.04%) | |
| **Place in the sibling (7 y)** | | | | | | | | |
| Single child | 115 (6.41%) | 103 (7.43%) | 76 (8.66%) | 0.024 | 119 (6.52%) | 107 (8.66%) | 79 (7.92%) | 0.344 |
| Eldest | 511 (28.50%) | 347 (25.04%) | 210 (23.92%) | | 481 (26.37%) | 317 (25.67%) | 273 (27.38%) | |
| ≥ 2 | 961 (53.60%) | 741 (53.46%) | 476 (54.21%) | | 1015 (55.65%) | 661 (53.52%) | 528 (52.96%) | |
| Missing | 206 (11.49%) | 195 (14.07%) | 116 (13.21%) | | 209 (11.46%) | 150 (12.15%) | 117 (11.74%) | |
| **ACEs (7-16y)** | | | | | | | | |
| None | 1285 (71.67%) | 891 (64.29%) | 545 (62.07%) | <0.001 | 1322 (72.48%) | 827 (66.96%) | 626 (62.79%) | <0.001 |
| One or more | 399 (22.25%) | 374 (26.98%) | 259 (29.50%) | | 389 (21.33%) | 320 (25.91%) | 298 (29.89%) | |
| Missing | 109 (6.08%) | 121 (8.73%) | 74 (8.43%) | | 113 (6.20%) | 88 (7.13%) | 73 (7.32%) | |
| **Health problems in childhood (7-16y)** | | | | | | | | |
| No | 1360 (75.85%) | 1002 (72.29%) | 626 (71.30%) | 0.014 | 1431 (78.45%) | 923 (74.74%) | 748 (75.03%) | 0.090 |

(*Continued*)

**Table 1.** (Continued)

| | Allostatic load at 44y | | | | | | | |
|---|---|---|---|---|---|---|---|---|
| | A. Men n(%) n = 4 075 (50%) | | | | B. Women n(%) n = 4 056 (50%) | | | |
| | Low n(%) | Medium n(%) | High n(%) | P-value* | Low n(%) | Medium n(%) | High n(%) | P-value* |
| | 1793 (44.20%) | 1386 (34.16%) | 878 (21.64%) | | 1824 (44.97%) | 1235 (30.45%) | 997 (24.58%) | |
| Yes | 416 (23.20%) | 376 (27.13%) | 248 (28.25%) | | 383 (21.00%) | 302 (24.45%) | 244 (24.47%) | |
| Missing | 17 (0.95%) | 8 (0.58%) | 4 (0.46%) | | 10 (0.55%) | 10 (0.81%) | 5 (0.50%) | |
| **Copy-a-Design test (7y)** | | | | | | | | |
| Score: med [p25-p75] | 8[6–9] | 7[6–9] | 7[6–8] | <0.001 | 8[6–9] | 7[6–8] | 7[6–8] | <0.001 |
| Missing | 183 (10.21%) | 180 (12.99%) | 110 (12.53%) | | 188 (10.31%) | 131 (10.61%) | 109 (10.93%) | |
| *Intermediate life course variables* | | | | | | | | |
| **Education level (23y)** | | | | | | | | |
| A level | 468 (26.10%) | 270 (19.48%) | 117 (13.33%) | <0.001 | 465 (25.49%) | 215 (17.41%) | 137 (13.74%) | <0.001 |
| O level | 601 (33.52%) | 453 (32.68%) | 270 (30.75%) | | 746 (40.90%) | 498 (40.32%) | 354 (35.51%) | |
| No level | 471 (26.27%) | 456 (32.90%) | 353 (40.21%) | | 409 (22.42%) | 366 (29.64%) | 365 (36.61%) | |
| Missing | 253 (14.11%) | 207 (14.94%) | 138 (15.72%) | | 204 (11.18%) | 156 (12.63%) | 141 (14.14%) | |
| **Malaise inventory (23y)** | | | | | | | | |
| No psychological distress | 1497 (83.49%) | 1134 (81.82%) | 705 (80.30%) | 0.156 | 1508 (82.68%) | 951 (77.00%) | 737 (73.92%) | <0.001 |
| Psychological distress | 42 (2.34%) | 45 (3.25%) | 33 (3.76%) | | 109 (5.98%) | 127 (10.28%) | 121 (12.14%) | |
| Missing | 254 (14.17%) | 207 (14.94%) | 140 (15.95%) | | 207 (11.35%) | 157 (12.71%) | 139 (13.94%) | |
| **Sense of personal control (33y)** | | | | | | | | |
| Internal | 1387 (77.36%) | 1018 (73.45%) | 628 (71.53%) | 0.012 | 1440 (78.95%) | 960 (77.73%) | 740 (74.22%) | 0.037 |
| External | 132 (7.36%) | 114 (8.23%) | 78 (8.88%) | | 187 (10.25%) | 129 (10.45%) | 135 (13.54%) | |
| Missing | 274 (15.28%) | 254 (18.33%) | 172 (19.59%) | | 197 (10.80%) | 146 (11.82%) | 122 (12.24%) | |
| **Occupational social class (33y)** | | | | | | | | |
| I & II | 713 (39.77%) | 460 (33.19%) | 245 (27.90%) | <0.001 | 609 (33.39%) | 339 (27.45%) | 225 (22.57%) | <0.001 |
| IIINM | 178 (9.93%) | 113 (8.15%) | 68 (7.74%) | | 553 (30.32%) | 393 (31.82%) | 306 (30.69%) | |
| IIIM | 442 (24.65%) | 396 (28.57%) | 265 (30.18%) | | 108 (5.92%) | 77 (6.23%) | 70 (7.02%) | |
| IV&V | 186 (10.37%) | 163 (11.76%) | 134 (15.26%) | | 289 (15.84%) | 242 (19.60%) | 217 (21.77%) | |
| Missing | 274 (15.28%) | 254 (18.33%) | 166 (18.91%) | | 265 (14.53%) | 184 (14.90%) | 179 (17.95%) | |
| **Wealth (33y)** | | | | | | | | |
| Owner highest price | 370 (20.64%) | 203 (14.65%) | 87 (9.91%) | <0.001 | 395 (21.66%) | 186 (15.06%) | 118 (11.84%) | <0.001 |
| Owner high price | 356 (19.85%) | 220 (15.87%) | 109 (12.41%) | | 350 (19.19%) | 203 (16.44%) | 129 (12.94%) | |
| Owner median price | 293 (16.34%) | 232 (16.74%) | 141 (16.06%) | | 329 (18.04%) | 234 (18.95%) | 156 (15.65%) | |
| Owner lowest price | 281 (15.67%) | 224 (16.16%) | 169 (19.25%) | | 281 (15.41%) | 200 (16.19%) | 182 (18.25%) | |
| Not owner | 273 (15.23%) | 293 (21.14%) | 231 (26.31%) | | 274 (15.02%) | 270 (21.86%) | 299 (29.99%) | |
| Missing | 220 (12.27%) | 214 (15.44%) | 141 (16.06%) | | 195 (10.69%) | 142 (11.50%) | 113 (11.33%) | |
| **Smoking (42y)** | | | | | | | | |
| Non-smoker | 894 (49.86%) | 572 (41.27%) | 297 (33.83%) | <0.001 | 885 (48.52%) | 530 (42.91%) | 398 (39.92%) | <0.001 |
| Ex smoker | 510 (28.44%) | 328 (23.67%) | 200 (22.78%) | | 506 (27.74%) | 295 (23.89%) | 184 (18.46%) | |
| Smoker < 10 cig./day | 130 (7.25%) | 108 (7.79%) | 51 (5.81%) | | 143 (7.84%) | 85 (6.88%) | 65 (6.52%) | |
| Smoker 10 to 19 cig./day | 90 (5.02%) | 116 (8.37%) | 99 (11.28%) | | 134 (7.35%) | 166 (13.44%) | 118 (11.84%) | |
| Smoker more than 20 cig./day | 116 (6.47%) | 203 (14.65%) | 200 (22.78%) | | 106 (5.81%) | 129 (10.45%) | 193 (19.36%) | |
| Missing | 53 (2.96%) | 59 (4.26%) | 31 (3.53%) | | 50 (2.74%) | 30 (2.43%) | 39 (3.91%) | |
| **Alcohol consumption (42y)** | | | | | | | | |
| Moderate | 1078 (60.12%) | 730 (52.67%) | 375 (42.71%) | <0.001 | 1249 (68.48%) | 769 (62.27%) | 542 (54.36%) | <0.001 |
| Abstinent | 237 (13.22%) | 233 (16.81%) | 192 (21.87%) | | 416 (22.81%) | 364 (29.47%) | 346 (34.70%) | |
| High | 425 (23.70%) | 365 (26.33%) | 280 (31.89%) | | 109 (5.98%) | 72 (5.83%) | 70 (7.02%) | |
| Missing | 53 (2.96%) | 58 (4.18%) | 31 (3.53%) | | 50 (2.74%) | 30 (2.43%) | 39 (3.91%) | |

*(Continued)*

**Table 1.** (Continued)

| | Allostatic load at 44y | | | | | | | |
|---|---|---|---|---|---|---|---|---|
| | **A. Men n(%) n = 4 075 (50%)** | | | | **B. Women n(%) n = 4 056 (50%)** | | | |
| | Low n(%) | Medium n(%) | High n(%) | *P-value** | Low n(%) | Medium n(%) | High n(%) | *P-value** |
| | 1793 (44.20%) | 1386 (34.16%) | 878 (21.64%) | | 1824 (44.97%) | 1235 (30.45%) | 997 (24.58%) | |
| **Physical activity (42y)** | | | | | | | | |
| Active | 1253 (69.88%) | 877 (63.28%) | 487 (55.47%) | <0.001 | 1252 (68.64%) | 793 (64.21%) | 557 (55.87%) | <0.001 |
| Moderate | 168 (9.37%) | 122 (8.80%) | 90 (10.25%) | | 124 (6.80%) | 103 (8.34%) | 71 (7.12%) | |
| Inactive | 318 (17.74%) | 329 (23.74%) | 270 (30.75%) | | 398 (21.82%) | 308 (24.94%) | 330 (33.10%) | |
| Missing | 54 (3.01%) | 58 (4.18%) | 31 (3.53%) | | 50 (2.74%) | 31 (2.51%) | 39 (3.91%) | |

The multivariable results of the association between parental interest and allostatic load, examining the *a priori* set of confounding and intermediate factors are presented in Tables 3 and 4, for men and women respectively. Men with parents perceived as uninterested or not very interested in their child's education had higher allostatic load scores at 44 years compared to those perceived as interested by the school teacher (Model 1, β = 0.41 [0.29; 0.54]). After adjustment for prior confounders, the link between parental interest and allostatic load was weakened (Model 2, β = 0.18 [0.03; 0.32]) partly attributable to parental SEC, ACEs, health problems in childhood and cognitive skills. The associations between parental interest and allostatic load were rendered insignificant after including educational attainment in Model 3 (Model 3, β = 0.06 [-0.09; 0.21]). Further adjusting for psychological status at age 23 (Model 4) and occupational social class and wealth (Model 5) and health behaviors (Model 6) did not change the result patterns.

A similar pattern was observed for women: women whose parents were perceived as uninterested or not very interested by the teacher had higher allostatic load scores at 44 years (Model 1, β = 0.69 [0.54; 0.83]). The association between parental interest and allostatic load was attenuated after controlling for early life confounder (Model 2, β = 0.40 [0.24; 0.56]) partly explained by parental SEC, mother's level of education and ACEs and cognitive skills. Further adjustment for educational attainment reduced the strength of the association (Model 3, β = 0.28 [0.10; 0.45]). When psychological status was accounted for, the association between parental interest and allostatic load was marginally affected (Model 4, β = 0,26 [0.08; 0.43]) but was explained by the malaise inventory. Further adjustment for occupational social class and wealth lightly attenuated the association with income affecting it more strongly (Model 5, β = 0.21 [0.04; 0.39]) as well as health behaviors (Model 6, β = 0.18 [0.01; 0.35]). When all potential mediators were controlled for, parental interest remained significantly associated with allostatic load score (Model 6, β = 0.18 [0.01; 0.35]).

The analyses of the direct and indirect effects of parental interest on allostatic load are presented in Figs 2 and 3. For men, the direct link between parental interest and allostatic load was completely mediated, mainly by the educational pathway (67% of the total indirect effect) but also through other intermediate factors (33% of the total indirect effect).

For women, 55% of the link between parental interest and allostatic load was mediated, through the educational pathway (30% of the total indirect effect) and by other intermediate factors (25% of the total indirect effect). A direct effect of 45% persisted after adjustment for confounding factors and mediators.

For this calculation among men, we did not consider the direct effect parental interest on allostatic load because the estimation $\hat{\beta}_{PI.2} - 0.01[-0.16; \ 0.14]$ had a non-significant value close to 0.

**Table 2. Bivariate statistics on the complete-case sample (n = 8 113): Parental interest categories according to confounding and intermediate variables in (A) men and in (B) women.**

| | Parental interest (7-16y) | | | | | | | |
|---|---|---|---|---|---|---|---|---|
| | A. Men n(%) n = 4 075 (50%) | | | | B. Women n(%) n = 4 056 (50%) | | | |
| | Both interested n (%) | Low/No interest n (%) | Missing n (%) | P-value* | Both interested n (%) | Low/No interest n (%) | Missing n (%) | P-value* |
| | 1912 (47.13%) | 1703 (41.98%) | 442 (10.89%) | | 1906 (46.99%) | 1666 (41.07%) | 484 (11.93%) | |
| **Allostatic load (44y)** | | | | | | | | |
| Low | 930 (48.64%) | 698 (40.99%) | 165 (37.33%) | <0.001 | 989 (51.89%) | 654 (39.26%) | 181 (37.40%) | <0.001 |
| Medium | 634 (33.16%) | 578 (33.94%) | 174 (39.37%) | | 555 (29.12%) | 524 (31.45%) | 156 (32.23%) | |
| High | 348 (18.20%) | 427 (25.07%) | 103 (23.30%) | | 362 (18.99%) | 488 (29.29%) | 147 (30.37%) | |
| *Prior confounders* | | | | | | | | |
| **Parental SEC (birth)** | | | | | | | | |
| I & II | 564 (29.50%) | 135 (7.93%) | 49 (11.09%) | <0.001 | 526 (27.60%) | 132 (7.92%) | 48 (9.92%) | <0.001 |
| IIINM | 223 (11.66%) | 128 (7.52%) | 29 (6.56%) | | 238 (12.49%) | 119 (7.14%) | 34 (7.02%) | |
| IIIM | 801 (41.89%) | 905 (53.14%) | 207 (46.83%) | | 788 (41.34%) | 876 (52.58%) | 247 (51.03%) | |
| IV&V | 229 (11.98%) | 441 (25.90%) | 98 (22.17%) | | 250 (13.12%) | 445 (26.71%) | 102 (21.07%) | |
| Missing | 95 (4.97%) | 94 (5.52%) | 59 (13.35%) | | 104 (5.46%) | 94 (5.64%) | 53 (10.95%) | |
| **Material living conditions (7y)** | | | | | | | | |
| Advantaged | 1375 (71.91%) | 930 (54.61%) | 190 (42.99%) | <0.001 | 1342 (70.41%) | 902 (54.14%) | 219 (45.25%) | <0.001 |
| Disadvantaged | 292 (15.27%) | 528 (31.00%) | 118 (26.70%) | | 315 (16.53%) | 548 (32.89%) | 144 (29.75%) | |
| Missing | 245 (12.81%) | 245 (14.39%) | 134 (30.32%) | | 249 (13.06%) | 216 (12.97%) | 121 (25.00%) | |
| **Father's level of education (7y)** | | | | | | | | |
| Left school ≥15y | 655 (34.26%) | 178 (10.45%) | 40 (9.05%) | <0.001 | 657 (34.47%) | 198 (11.88%) | 51 (10.54%) | <0.001 |
| Left school <14y | 1056 (55.23%) | 1268 (74.46%) | 250 (56.56%) | | 1023 (53.67%) | 1237 (74.25%) | 279 (57.64%) | |
| Missing | 201 (10.51%) | 257 (15.09%) | 152 (34.39%) | | 226 (11.86%) | 231 (13.87%) | 154 (31.82%) | |
| **Mother's level of education (birth)** | | | | | | | | |
| Left school ≥15y | 705 (36.87%) | 244 (14.33%) | 74 (16.74%) | <0.001 | 725 (38.04%) | 218 (13.09%) | 82 (16.94%) | <0.001 |
| Left school <14y | 1111 (58.11%) | 1376 (80.80%) | 313 (70.81%) | | 1084 (56.87%) | 1359 (81.57%) | 353 (72.93%) | |
| Missing | 96 (5.02%) | 83 (4.87%) | 55 (12.44%) | | 97 (5.09%) | 89 (5.34%) | 49 (10.12%) | |
| **Reading activities (7y)** | | | | | | | | |
| Every week | 1126 (58.89%) | 747 (43.86%) | 157 (35.52%) | <0.001 | 1076 (56.45%) | 715 (42.92%) | 175 (36.16%) | <0.001 |
| Occasionally | 459 (24.01%) | 514 (30.18%) | 110 (24.89%) | | 509 (26.71%) | 586 (35.17%) | 146 (30.17%) | |
| Hardly ever | 150 (7.85%) | 224 (13.15%) | 35 (7.92%) | | 134 (7.03%) | 183 (10.98%) | 39 (8.06%) | |
| Missing | 177 (9.26%) | 218 (12.80%) | 140 (31.67%) | | 187 (9.81%) | 182 (10.92%) | 124 (25.62%) | |
| **Outdoor activities (7y)** | | | | | | | | |
| Most weeks | 1615 (84.47%) | 1202 (70.58%) | 259 (58.60%) | <0.001 | 1599 (83.89%) | 1257 (75.45%) | 316 (65.29%) | <0.001 |

(*Continued*)

**Table 2.** (*Continued*)

| | Parental interest (7-16y) | | | | | | | |
|---|---|---|---|---|---|---|---|---|
| | A. Men n(%) n = 4 075 (50%) | | | | B. Women n(%) n = 4 056 (50%) | | | |
| | Both interested n (%) | Low/No interest n (%) | Missing n (%) | *P-value*[*] | Both interested n (%) | Low/No interest n (%) | Missing n (%) | *P-value*[*] |
| | 1912 (47.13%) | 1703 (41.98%) | 442 (10.89%) | | 1906 (46.99%) | 1666 (41.07%) | 484 (11.93%) | |
| Occasionally/hardly ever | 125 (6.54%) | 283 (16.62%) | 44 (9.95%) | | 128 (6.72%) | 223 (13.39%) | 45 (9.30%) | |
| Missing | 172 (9.00%) | 218 (12.80%) | 139 (31.45%) | | 179 (9.39%) | 186 (11.16%) | 123 (25.41%) | |
| **Place in the sibling (7 y)** | | | | | | | | |
| Single child | 155 (8.11%) | 99 (5.81%) | 40 (9.05%) | <**0.001** | 172 (9.02%) | 89 (5.34%) | 44 (9.09%) | <**0.001** |
| Eldest | 612 (32.01%) | 377 (22.14%) | 79 (17.87%) | | 596 (31.27%) | 377 (22.63%) | 98 (20.25%) | |
| ≥ 2 | 975 (50.99%) | 1017 (59.72%) | 186 (42.08%) | | 965 (50.63%) | 1021 (61.28%) | 218 (45.04%) | |
| Missing | 170 (8.89%) | 210 (12.33%) | 137 (31.00%) | | 173 (9.08%) | 179 (10.74%) | 124 (25.62%) | |
| **ACEs (7-16y)** | | | | | | | | |
| None | 1534 (80.23%) | 1022 (60.01%) | 165 (37.33%) | <**0.001** | 1531 (80.33%) | 1037 (62.24%) | 207 (42.77%) | <**0.001** |
| One or more | 287 (15.01%) | 566 (33.24%) | 179 (40.50%) | | 267 (14.01%) | 541 (32.47%) | 199 (41.12%) | |
| Missing | 91 (4.76%) | 115 (6.75%) | 98 (22.17%) | | 108 (5.67%) | 88 (5.28%) | 78 (16.12%) | |
| **Health problems in childhood (7-16y)** | | | | | | | | |
| No | 1415 (74.01%) | 1260 (73.99%) | 313 (70.81%) | <**0.001** | 1495 (78.44%) | 1254 (75.27%) | 353 (72.93%) | <**0.001** |
| Yes | 494 (25.84%) | 435 (25.54%) | 111 (25.11%) | | 409 (21.46%) | 406 (24.37%) | 114 (23.55%) | |
| Missing | 3 (0.16%) | 8 (0.47%) | 18 (4.07%) | | 2 (0.10%) | 6 (0.36%) | 17 (3.51%) | |
| **Copy-a-Design test (7y)** | | | | | | | | |
| Score: med [p25-p75] | 8[6–9] | 7[6–8] | 7[6–8] | <**0.001** | 8[6–9] | 7[6–8] | 7[6–8] | <**0.001** |
| Missing | 145 (30.66%) | 183 (38.69%) | 145 (30.66%) | | 151 (35.28%) | 157 (36.68%) | 120 (28.04%) | |
| *Intermediate life course variables* | | | | | | | | |
| **Education level (23y)** | | | | | | | | |
| A level | 685 (35.83%) | 120 (7.05%) | 50 (11.31%) | <**0.001** | 662 (34.73%) | 96 (5.76%) | 59 (12.19%) | <**0.001** |
| O level | 687 (35.93%) | 515 (30.24%) | 122 (27.60%) | | 811 (42.55%) | 621 (37.27%) | 166 (34.30%) | |
| No level | 289 (15.12%) | 807 (47.39%) | 184 (41.63%) | | 239 (12.54%) | 724 (43.46%) | 177 (36.57%) | |
| Missing | 251 (13.13%) | 261 (15.33%) | 86 (19.46%) | | 194 (10.18%) | 225 (13.51%) | 82 (16.94%) | |
| **Malaise inventory (23y)** | | | | | | | | |
| No psychological distress | 1623 (84.88%) | 1374 (80.68%) | 339 (76.70%) | <**0.001** | 1610 (84.47%) | 1237 (74.25%) | 349 (72.11%) | <**0.001** |
| Psychological distress | 36 (1.88%) | 67 (3.93%) | 17 (3.85%) | | 103 (5.40%) | 201 (12.06%) | 53 (10.95%) | |
| Missing | 253 (13.23%) | 262 (15.38%) | 86 (19.46%) | | 193 (10.13%) | 228 (13.69%) | 82 (16.94%) | |
| **Sense of personal control (33y)** | | | | | | | | |
| Internal | 1513 (79.13%) | 1218 (71.52%) | 302 (68.33%) | <**0.001** | 1581 (82.95%) | 1222 (73.35%) | 337 (69.63%) | <**0.001** |
| External | 116 (6.07%) | 169 (9.92%) | 39 (8.82%) | | 146 (7.66%) | 231 (13.87%) | 74 (15.29%) | |

(*Continued*)

**Table 2.** (Continued)

| | Parental interest (7-16y) | | | | | | | |
|---|---|---|---|---|---|---|---|---|
| | A. Men n(%) n = 4 075 (50%) | | | | B. Women n(%) n = 4 056 (50%) | | | |
| | Both interested n (%) | Low/No interest n (%) | Missing n (%) | P-value* | Both interested n (%) | Low/No interest n (%) | Missing n (%) | P-value* |
| | 1912 (47.13%) | 1703 (41.98%) | 442 (10.89%) | | 1906 (46.99%) | 1666 (41.07%) | 484 (11.93%) | |
| Missing | 283 (14.80%) | 316 (18.56%) | 101 (22.85%) | | 179 (9.39%) | 213 (12.79%) | 73 (15.08%) | |
| **Occupational social class (33y)** | | | | | | | | |
| I & II | 926 (48.43%) | 378 (22.20%) | 114 (25.79%) | <**0.001** | 745 (39.09%) | 303 (18.19%) | 125 (25.83%) | <**0.001** |
| IIINM | 207 (10.83%) | 119 (6.99%) | 33 (7.47%) | | 593 (31.11%) | 505 (30.31%) | 154 (31.82%) | |
| IIIM | 363 (18.99%) | 608 (35.70%) | 132 (29.86%) | | 98 (5.14%) | 122 (7.32%) | 35 (7.23%) | |
| IV&V | 123 (6.43%) | 288 (16.91%) | 72 (16.29%) | | 220 (11.54%) | 439 (26.35%) | 89 (18.39%) | |
| Missing | 293 (15.32%) | 310 (18.20%) | 91 (20.59%) | | 250 (13.12%) | 297 (17.83%) | 81 (16.74%) | |
| **Wealth (33y)** | | | | | | | | |
| Owner highest price | 422 (22.07%) | 178 (10.45%) | 60 (13.57%) | <**0.001** | 447 (23.45%) | 189 (11.34%) | 63 (13.02%) | <**0.001** |
| Owner high price | 400 (20.92%) | 223 (13.09%) | 62 (14.03%) | | 401 (21.04%) | 219 (13.15%) | 62 (12.81%) | |
| Owner median price | 318 (16.63%) | 287 (16.85%) | 61 (13.80%) | | 366 (19.20%) | 261 (15.67%) | 92 (19.01%) | |
| Owner lowest price | 250 (13.08%) | 351 (20.61%) | 73 (16.52%) | | 239 (12.54%) | 335 (20.11%) | 89 (18.39%) | |
| Not owner | 287 (15.01%) | 404 (23.72%) | 106 (23.98%) | | 268 (14.06%) | 458 (27.49%) | 117 (24.17%) | |
| Missing | 235 (12.29%) | 260 (15.27%) | 80 (18.10%) | | 185 (9.71%) | 204 (12.24%) | 61 (12.60%) | |
| **Smoking (42y)** | | | | | | | | |
| Non-smoker | 952 (49.79%) | 631 (37.05%) | 180 (40.72%) | <**0.001** | 1000 (52.47%) | 619 (37.15%) | 194 (40.08%) | <**0.001** |
| Ex smoker | 468 (24.48%) | 457 (26.83%) | 113 (25.57%) | | 480 (25.18%) | 384 (23.05%) | 121 (25.00%) | |
| Smoker < 10 cig./day | 154 (8.05%) | 107 (6.28%) | 28 (6.33%) | | 133 (6.98%) | 134 (8.04%) | 26 (5.37%) | |
| Smoker 10 to 19 cig./day | 113 (5.91%) | 159 (9.34%) | 33 (7.47%) | | 138 (7.24%) | 228 (13.69%) | 52 (10.74%) | |
| Smoker more than 20 cig./day | 168 (8.79%) | 286 (16.79%) | 65 (14.71%) | | 106 (5.56%) | 249 (14.95%) | 73 (15.08%) | |
| Missing | 57 (2.98%) | 63 (3.70%) | 23 (5.20%) | | 49 (2.57%) | 52 (3.12%) | 18 (3.72%) | |
| **Alcohol consumption (42y)** | | | | | | | | |
| Moderate | 1143 (59.78%) | 821 (48.21%) | 219 (49.55%) | <**0.001** | 1312 (68.84%) | 975 (58.52%) | 273 (56.40%) | <**0.001** |
| Abstinent | 232 (12.13%) | 320 (18.79%) | 110 (24.89%) | | 421 (22.09%) | 538 (32.29%) | 167 (34.50%) | |
| High | 480 (25.10%) | 500 (29.36%) | 90 (20.36%) | | 124 (6.51%) | 101 (6.06%) | 26 (5.37%) | |
| Missing | 57 (2.98%) | 62 (3.64%) | 23 (5.20%) | | 49 (2.57%) | 52 (3.12%) | 18 (3.72%) | |
| **Physical activity (42y)** | | | | | | | | |
| Active | 1305 (68.25%) | 1036 (60.83%) | 276 (62.44%) | <**0.001** | 1282 (67.26%) | 1026 (61.58%) | 294 (60.74%) | <**0.001** |
| Moderate | 207 (10.83%) | 134 (7.87%) | 39 (8.82%) | | 155 (8.13%) | 112 (6.72%) | 31 (6.40%) | |
| Inactive | 342 (17.89%) | 471 (27.66%) | 104 (23.53%) | | 420 (22.04%) | 475 (28.51%) | 141 (29.13%) | |
| Missing | 58 (3.03%) | 62 (3.64%) | 23 (5.20%) | | 49 (2.57%) | 53 (3.18%) | 18 (3.72%) | |

Abbreviations and symbols: n = number of people; med = median; p25 = 25e percentile; p75 = 75e percentile; statistically significant results at the 5% threshold are in bold. Values corresponding to the categories of allostatic load: Low: [0–2]; Medium: [3–4]; High: [5–12].

*P-values were calculated using the chi-squared test for categorical variables and Wilcoxon signed-rank test for the continuous variable.

**Table 3. Life course multivariable linear regression between allostatic load and parental interest using data obtained from multiple imputation for men (n = 4 057).**

| | Model 1 | | Model 2 | | Model 3 | | Model 4 | | Model 5 | | Model 6 | |
|---|---|---|---|---|---|---|---|---|---|---|---|---|
| | Coeff. [CI 95%] | P-value | Coeff. [CI 95%] | P-value | Coeff. [CI 95%] | P-value | Coeff. [CI 95%] | P-value | Coeff. [CI 95%] | P-value | Coeff. [CI 95%] | P-value |
| **Parental interest** | | | | | | | | | | | | |
| Both interested | ref | | ref | | ref | | ref | | ref | | ref | |
| Low/No interest | 0.41 [0.29; 0.54] | <0.001 | 0.18 [0.03; 0.32] | 0.015 | 0.06 [-0.09; 0.21] | 0.439 | 0.05 [-0.10; 0.21] | 0.475 | 0.02 [-0.13; 0.17] | 0.823 | -0.01 [-0.16; 0.14] | 0.866 |
| **Parental SEC (birth)** | | | | | | | | | | | | |
| I & II | | | ref | | ref | | ref | | ref | | ref | |
| IIINM | | | 0.04 [-0.2; 0.28] | 0.732 | 0.02 [-0.22; 0.26] | 0.867 | 0.02 [-0.22; 0.26] | 0.845 | 0.05 [-0.19; 0.29] | 0.693 | 0.02 [-0.21; 0.26] | 0.836 |
| IIIM | | | 0.32 [0.12; 0.51] | 0.001 | 0.26 [0.06; 0.45] | 0.009 | 0.26 [0.07; 0.46] | 0.008 | 0.26 [0.07; 0.45] | 0.008 | 0.22 [0.03; 0.41] | 0.022 |
| IV&V | | | 0.40 [0.17; 0.63] | 0.001 | 0.32 [0.09; 0.55] | 0.006 | 0.32 [0.10; 0.55] | 0.005 | 0.29 [0.06; 0.51] | 0.013 | 0.25 [0.03; 0.47] | 0.029 |
| **Material living conditions (7y)** | | | | | | | | | | | | |
| Advantaged | | | ref | | ref | | ref | | ref | | ref | |
| Disadvantaged | | | 0.13 [-0.04; 0.30] | 0.121 | 0.13 [-0.04; 0.29] | 0.132 | 0.13 [-0.04; 0.29] | 0.139 | 0.10 [-0.07; 0.27] | 0.227 | 0.10 [-0.06; 0.26] | 0.236 |
| **Father's level of education (7y)** | | | | | | | | | | | | |
| Left school ≥15y | | | ref | | ref | | ref | | ref | | ref | |
| Left school <14y | | | 0.08 [-0.09; 0.26] | 0.366 | 0.01 [-0.16; 0.19] | 0.872 | 0.01 [-0.16; 0.19] | 0.875 | 0.002 [-0.17; 0.18] | 0.983 | 0.04 [-0.14; 0.21] | 0.689 |
| **Mother's level of education (birth)** | | | | | | | | | | | | |
| Left school ≥15y | | | ref | | ref | | ref | | ref | | ref | |
| Left school <14y | | | 0.11 [-0.04; 0.27] | 0.156 | 0.06 [-0.10; 0.22] | 0.448 | 0.06 [-0.09; 0.22] | 0.443 | 0.05 [-0.11; 0.20] | 0.572 | 0.06 [-0.10; 0.21] | 0.465 |
| **Reading activities (7y)** | | | | | | | | | | | | |
| Every week | | | ref | | ref | | ref | | ref | | ref | |
| Occasionally | | | -0.02 [-0.17; 0.13] | 0.771 | -0.04 [-0.19; 0.11] | 0.575 | -0.04 [-0.19; 0.11] | 0.584 | -0.03 [-0.18; 0.12] | 0.732 | -0.02 [-0.17; 0.12] | 0.770 |
| Hardly ever | | | 0.08 [-0.15; 0.31] | 0.475 | 0.08 [-0.15; 0.31] | 0.502 | 0.08 [-0.15; 0.31] | 0.506 | 0.09 [-0.14; 0.32] | 0.454 | 0.06 [-0.16; 0.29] | 0.597 |
| **Outdoor activities (7y)** | | | | | | | | | | | | |
| Most weeks | | | ref | | ref | | ref | | ref | | ref | |
| Occasionally/hardly ever | | | -0.12 [-0.33; 0.09] | 0.261 | -0.13 [-0.34; 0.07] | 0.209 | -0.13 [-0.34; 0.07] | 0.204 | -0.15 [-0.36; 0.06] | 0.151 | -0.14 [-0.34; 0.06] | 0.169 |
| **Place in the sibling (7 y)** | | | | | | | | | | | | |
| Single child | | | ref | | ref | | ref | | ref | | ref | |
| Elder | | | -0.19 [-0.44; 0.07] | 0.152 | -0.20 [-0.46; 0.05] | 0.118 | -0.20 [-0.45; 0.05] | 0.123 | -0.2 [-0.45; 0.05] | 0.119 | -0.19 [-0.44; 0.06] | 0.139 |
| ≥ 2 | | | -0.15 [-0.38; 0.09] | 0.215 | -0.19 [-0.42; 0.05] | 0.119 | -0.19 [-0.42; 0.05] | 0.117 | -0.2 [-0.43; 0.03] | 0.095 | -0.20 [-0.43; 0.03] | 0.092 |
| **ACEs (7-16y)** | | | | | | | | | | | | |
| None | | | ref | | ref | | ref | | ref | | ref | |
| One or more | | | 0.20 [0.06; 0.34] | 0.005 | 0.17 [0.03; 0.31] | 0.021 | 0.16 [0.02; 0.30] | 0.026 | 0.13 [-0.01; 0.27] | 0.068 | 0.06 [-0.08; 0.20] | 0.432 |
| **Health problems in childhood (7-16y)** | | | | | | | | | | | | |
| No | | | ref | | ref | | ref | | ref | | ref | |
| Yes | | | 0.19 [0.06; 0.33] | 0.006 | 0.18 [0.04; 0.32] | 0.010 | 0.17 [0.04; 0.31] | 0.013 | 0.14 [0.002; 0.28] | 0.046 | 0.13 [-0.005; 0.26] | 0.059 |
| **Copy-a-Design test (7y)** | | | | | | | | | | | | |
| Score: med [p25-p75] | | | -0.06 [-0.09; -0.02] | 0.001 | -0.04 [-0.08; -0.01] | 0.022 | -0.04 [-0.08; -0.01] | 0.023 | -0.03 [-0.07; 0.0002] | 0.051 | -0.03 [-0.06; 0.004] | 0.081 |
| **Education level (23y)** | | | | | | | | | | | | |
| A level | | | | | ref | | ref | | ref | | ref | |
| O level | | | | | 0.26 [0.09; 0.43] | 0.002 | 0.26 [0.09; 0.43] | 0.003 | 0.21 [0.03; 0.39] | 0.022 | 0.12 [-0.05; 0.29] | 0.179 |
| No level | | | | | 0.54 [0.34; 0.74] | <0.001 | 0.53 [0.33; 0.73] | <0.001 | 0.41 [0.19; 0.63] | <0.001 | 0.24 [0.02; 0.46] | 0.029 |
| **Malaise inventory (23y)** | | | | | | | | | | | | |
| No psychological distress | | | | | | | | | | | ref | |

(Continued)

**Table 3.** (Continued)

| | Model 1 | | Model 2 | | Model 3 | | Model 4 | | Model 5 | | Model 6 | |
|---|---|---|---|---|---|---|---|---|---|---|---|---|
| | Coeff. [CI 95%] | P-value | Coeff. [CI 95%] | P-value | Coeff. [CI 95%] | P-value | Coeff. [CI 95%] | P-value | Coeff. [CI 95%] | P-value | Coeff. [CI 95%] | P-value |
| Psychological distress | | | | | | | 0.09 [-0.25; 0.43] | 0.610 | 0.06 [-0.28; 0.40] | 0.730 | -0.03 [-0.37; 0.31] | 0.851 |
| **Sense of personal control (33y)** | | | | | | | | | | | | |
| Internal | | | | | | | ref | | ref | | ref | |
| External | | | | | | | 0.13 [-0.09; 0.35] | 0.257 | 0.06 [-0.17; 0.29] | 0.605 | -0.02 [-0.24; 0.21] | 0.873 |
| **Occupational social class (33y)** | | | | | | | | | | | | |
| I & II | | | | | | | | | ref | | ref | |
| IIINM | | | | | | | | | -0.12 [-0.33; 0.09] | 0.253 | -0.11 [-0.32; 0.09] | 0.265 |
| IIIM | | | | | | | | | 0.01 [-0.17; 0.18] | 0.932 | -0.05 [-0.22; 0.12] | 0.571 |
| IV&V | | | | | | | | | 0.03 [-0.19; 0.24] | 0.818 | -0.04 [-0.26; 0.18] | 0.745 |
| **Wealth (33y)** | | | | | | | | | | | | |
| Owner highest price | | | | | | | | | ref | | ref | |
| Owner high price | | | | | | | | | 0.06 [-0.14; 0.26] | 0.546 | 0.06 [-0.14; 0.26] | 0.548 |
| Owner median price | | | | | | | | | **0.29 [0.06; 0.51]** | **0.012** | **0.23 [0.01; 0.45]** | **0.038** |
| Owner lowest price | | | | | | | | | **0.35 [0.12; 0.57]** | **0.003** | **0.24 [0.02; 0.46]** | **0.030** |
| Not owner | | | | | | | | | **0.65 [0.44; 0.86]** | **<0.001** | **0.45 [0.23; 0.66]** | **<0.001** |
| **Smoking (42y)** | | | | | | | | | | | | |
| Non-smoker | | | | | | | | | | | ref | |
| Ex smoker | | | | | | | | | | | 0.07 [-0.08; 0.22] | 0.338 |
| Smoker < 10 cig./day | | | | | | | | | | | 0.08 [-0.16; 0.31] | 0.532 |
| Smoker 10 to 19 cig./day | | | | | | | | | | | **0.68 [0.44; 0.92]** | **<0.001** |
| Smoker more than 20 cig./day | | | | | | | | | | | **0.93 [0.73; 1.12]** | **<0.001** |
| **Alcohol consumption (42y)** | | | | | | | | | | | | |
| Moderate | | | | | | | | | | | ref | |
| Abstinent | | | | | | | | | | | **0.33 [0.16; 0.5]** | **<0.001** |
| High | | | | | | | | | | | **0.25 [0.11; 0.39]** | **0.001** |
| **Physical activity (42y)** | | | | | | | | | | | | |
| Active | | | | | | | | | | | ref | |
| Moderate | | | | | | | | | | | 0.10 [-0.10; 0.30] | 0.334 |
| Inactive | | | | | | | | | | | **0.30 [0.16; 0.45]** | **<0.001** |

**Table 4. Life course multivariable linear regression between allostatic load and parental interest using data obtained from multiple imputation for women (n = 4 056).**

| | Model 1 | | Model 2 | | Model 3 | | Model 4 | | Model 5 | | Model 6 | |
|---|---|---|---|---|---|---|---|---|---|---|---|---|
| | Coeff. [CI 95%] | P-value | Coeff. [CI 95%] | P-value | Coeff. [CI 95%] | P-value | Coeff. [CI 95%] | P-value | Coeff. [CI 95%] | P-value | Coeff. [CI 95%] | P-value |
| **Parental interest** | | | | | | | | | | | | |
| Both interested | ref | | ref | | ref | | ref | | ref | | ref | |
| Low/No interest | **0.69 [0.54; 0.83]** | **<0.001** | **0.40 [0.24; 0.56]** | **<0.001** | **0.28 [0.10; 0.45]** | **0.002** | **0.26 [0.08; 0.43]** | **0.004** | **0.21 [0.04; 0.39]** | **0.015** | **0.18 [0.01; 0.35]** | **0.041** |
| **Parental SEC (birth)** | | | | | | | | | | | | |
| I & II | | | ref | | ref | | ref | | ref | | ref | |
| IIINM | | | 0.17 [-0.10; 0.44] | 0.219 | 0.16 [-0.12; 0.43] | 0.262 | 0.16 [-0.11; 0.43] | 0.242 | 0.17 [-0.10; 0.43] | 0.224 | 0.14 [-0.13; 0.40] | 0.316 |
| IIIM | | | **0.42 [0.21; 0.63]** | **<0.001** | **0.38 [0.17; 0.59]** | **<0.001** | **0.38 [0.17; 0.59]** | **0.001** | **0.34 [0.13; 0.56]** | **0.001** | **0.32 [0.11; 0.53]** | **0.003** |
| IV&V | | | **0.50 [0.26; 0.75]** | **<0.001** | **0.44 [0.19; 0.69]** | **<0.001** | **0.44 [0.19; 0.68]** | **0.001** | **0.37 [0.12; 0.61]** | **0.004** | **0.32 [0.08; 0.57]** | **0.009** |
| **Material living conditions (7y)** | | | | | | | | | | | | |
| Advantaged | | | ref | | ref | | ref | | ref | | ref | |
| Disadvantaged | | | 0.07 [-0.09; 0.23] | 0.382 | 0.07 [-0.09; 0.23] | 0.407 | 0.06 [-0.10; 0.22] | 0.481 | 0.03 [-0.13; 0.19] | 0.706 | 0.001 [-0.16; 0.16] | 0.99 |
| **Father's level of education (7y)** | | | | | | | | | | | | |
| Left school ≥15y | | | ref | | ref | | ref | | ref | | ref | |
| Left school <14y | | | 0.07 [-0.11; 0.25] | 0.447 | 0.02 [-0.16; 0.20] | 0.801 | 0.02 [-0.16; 0.20] | 0.827 | -0.02 [-0.19; 0.16] | 0.856 | -0.03 [-0.21; 0.15] | 0.747 |
| **Mother's level of education (birth)** | | | | | | | | | | | | |
| Left school ≥15y | | | ref | | ref | | ref | | ref | | ref | |
| Left school <14y | | | **0.28 [0.11; 0.45]** | **0.001** | **0.23 [0.06; 0.41]** | **0.008** | **0.24 [0.06; 0.41]** | **0.007** | **0.23 [0.06; 0.41]** | **0.008** | **0.23 [0.06; 0.40]** | **0.008** |
| **Reading activities (7y)** | | | | | | | | | | | | |
| Every week | | | ref | | ref | | ref | | ref | | ref | |
| Occasionally | | | -0.12 [-0.27; 0.03] | 0.125 | -0.12 [-0.27; 0.03] | 0.116 | -0.12 [-0.27; 0.03] | 0.115 | -0.12 [-0.27; 0.03] | 0.123 | -0.13 [-0.28; 0.02] | 0.090 |
| Hardly ever | | | -0.07 [-0.30; 0.17] | 0.570 | -0.08 [-0.31; 0.16] | 0.523 | -0.08 [-0.31; 0.15] | 0.493 | -0.08 [-0.31; 0.15] | 0.491 | -0.11 [-0.34; 0.12] | 0.337 |
| **Outdoor activities (7y)** | | | | | | | | | | | | |
| Most weeks | | | ref | | ref | | ref | | ref | | ref | |
| Occasionally/hardly ever | | | 0.09 [-0.14; 0.33] | 0.432 | 0.08 [-0.15; 0.31] | 0.521 | 0.06 [-0.17; 0.29] | 0.594 | 0.04 [-0.18; 0.27] | 0.700 | 0.04 [-0.18; 0.27] | 0.703 |
| **Place in the sibling (7 y)** | | | | | | | | | | | | |
| Single child | | | ref | | ref | | ref | | ref | | ref | |
| Elder | | | -0.07 [-0.33; 0.20] | 0.625 | -0.08 [-0.35; 0.18] | 0.531 | -0.08 [-0.34; 0.18] | 0.538 | -0.08 [-0.34; 0.18] | 0.539 | -0.10 [-0.36; 0.15] | 0.422 |
| ≥ 2 | | | -0.20 [-0.47; 0.06] | 0.130 | -0.24 [-0.51; 0.02] | 0.072 | -0.25 [-0.51; 0.01] | 0.064 | -0.25 [-0.51; 0.01] | 0.063 | **-0.26 [-0.52; -0.004]** | **0.047** |
| **ACEs (7-16y)** | | | | | | | | | | | | |
| None | | | ref | | ref | | ref | | ref | | ref | |
| One or more | | | **0.19 [0.03; 0.35]** | **0.017** | **0.16 [0.002; 0.31]** | **0.046** | 0.14 [-0.012; 0.30] | 0.071 | 0.10 [-0.06; 0.26] | 0.219 | 0.05 [-0.11; 0.21] | 0.544 |

(*Continued*)

**Table 4.** (Continued)

| | Model 1 | | Model 2 | | Model 3 | | Model 4 | | Model 5 | | Model 6 | |
|---|---|---|---|---|---|---|---|---|---|---|---|---|
| | Coeff. [CI 95%] | P-value | Coeff. [CI 95%] | P-value | Coeff. [CI 95%] | P-value | Coeff. [CI 95%] | P-value | Coeff. [CI 95%] | P-value | Coeff. [CI 95%] | P-value |
| **Health problems in childhood (7-16y)** | | | | | | | | | | | | |
| No | | | ref | | ref | | ref | | ref | | ref | |
| Yes | | | 0.14 [-0.02; 0.29] | 0.086 | 0.12 [-0.04; 0.27] | 0.134 | 0.11 [-0.04; 0.26] | 0.164 | 0.10 [-0.05; 0.25] | 0.198 | 0.08 [-0.07; 0.23] | 0.300 |
| **Copy-a-Design test (7y)** | | | | | | | | | | | | |
| Score: med [p25-p75] | | | **-0.09 [-0.13; -0.05]** | **<0.001** | **-0.07 [-0.11; -0.03]** | **0.001** | **-0.06 [-0.10; -0.02]** | **0.002** | **-0.06 [-0.09; -0.02]** | **0.005** | **-0.05 [-0.09; -0.01]** | **0.008** |
| **Education level (23y)** | | | | | | | | | | | | |
| A level | | | | | ref | | ref | | ref | | ref | |
| O level | | | | | 0.15 [-0.05; 0.35] | 0.145 | 0.14 [-0.05; 0.34] | 0.154 | 0.08 [-0.14; 0.29] | 0.482 | 0.05 [-0.16; 0.26] | 0.658 |
| No level | | | | | **0.52 [0.28; 0.75]** | **<0.001** | **0.49 [0.26; 0.73]** | **<0.001** | **0.31 [0.05; 0.56]** | **0.019** | 0.20 [-0.06; 0.45] | 0.139 |
| **Malaise inventory (23y)** | | | | | | | | | | | | |
| No psychological distress | | | | | | | ref | | ref | | ref | |
| Psychological distress | | | | | | | **0.37 [0.13; 0.61]** | **0.003** | **0.35 [0.11; 0.59]** | **0.004** | **0.26 [0.02; 0.50]** | **0.032** |
| **Sense of personal control (33y)** | | | | | | | | | | | | |
| Internal | | | | | | | ref | | ref | | ref | |
| External | | | | | | | 0.09 [-0.14; 0.32] | 0.429 | -0.03 [-0.25; 0.20] | 0.819 | -0.10 [-0.32; 0.12] | 0.387 |
| **Occupational social class (33y)** | | | | | | | | | | | | |
| I & II | | | | | | | | | ref | | ref | |
| IIINM | | | | | | | | | 0.12 [-0.06; 0.31] | 0.182 | 0.14 [-0.04; 0.32] | 0.128 |
| IIIM | | | | | | | | | 0.08 [-0.21; 0.37] | 0.578 | 0.04 [-0.24; 0.32] | 0.784 |
| IV&V | | | | | | | | | 0.13 [-0.09; 0.35] | 0.245 | 0.11 [-0.11; 0.33] | 0.324 |
| **Wealth (33y)** | | | | | | | | | | | | |
| Owner highest price | | | | | | | | | ref | | ref | |
| Owner high price | | | | | | | | | 0.05 [-0.17; 0.27] | 0.635 | 0.05 [-0.16; 0.27] | 0.624 |
| Owner median price | | | | | | | | | 0.14 [-0.09; 0.37] | 0.236 | 0.12 [-0.10; 0.35] | 0.291 |
| Owner lowest price | | | | | | | | | **0.33 [0.10; 0.57]** | **0.006** | **0.25 [0.01; 0.48]** | **0.039** |
| Not owner | | | | | | | | | **0.74 [0.51; 0.97]** | **<0.001** | **0.55 [0.31; 0.78]** | **<0.001** |
| **Smoking (42y)** | | | | | | | | | | | | |
| Non-smoker | | | | | | | | | | | ref | |
| Ex smoker | | | | | | | | | | | **-0.20 [-0.36; -0.03]** | **0.018** |
| Smoker < 10 cig./day | | | | | | | | | | | -0.14 [-0.40; 0.12] | 0.286 |

(Continued)

**Table 4.** (Continued)

| | Model 1 | | Model 2 | | Model 3 | | Model 4 | | Model 5 | | Model 6 | |
|---|---|---|---|---|---|---|---|---|---|---|---|---|
| | Coeff. [CI 95%] | P-value | Coeff. [CI 95%] | P-value | Coeff. [CI 95%] | P-value | Coeff. [CI 95%] | P-value | Coeff. [CI 95%] | P-value | Coeff. [CI 95%] | P-value |
| Smoker 10 to 19 cig./day | | | | | | | | | | | **0.30 [0.07; 0.53]** | **0.009** |
| Smoker more than 20 cig./day | | | | | | | | | | | **0.86 [0.62; 1.09]** | **<0.001** |
| **Alcohol consumption (42y)** | | | | | | | | | | | | |
| Moderate | | | | | | | | | | | ref | |
| Abstinent | | | | | | | | | | | **0.31 [0.16; 0.46]** | **<0.001** |
| High | | | | | | | | | | | 0.05 [-0.21; 0.32] | 0.698 |
| **Physical activity (42y)** | | | | | | | | | | | | |
| Active | | | | | | | | | | | ref | |
| Moderate | | | | | | | | | | | 0.14 [-0.11; 0.39] | 0.267 |
| Inactive | | | | | | | | | | | **0.27 [0.12; 0.43]** | **0.001** |

## 4. Discussion

Parents' interest in their child's education as perceived by teachers measured when cohort members were school children, was associated with their physiological health in mid-life in

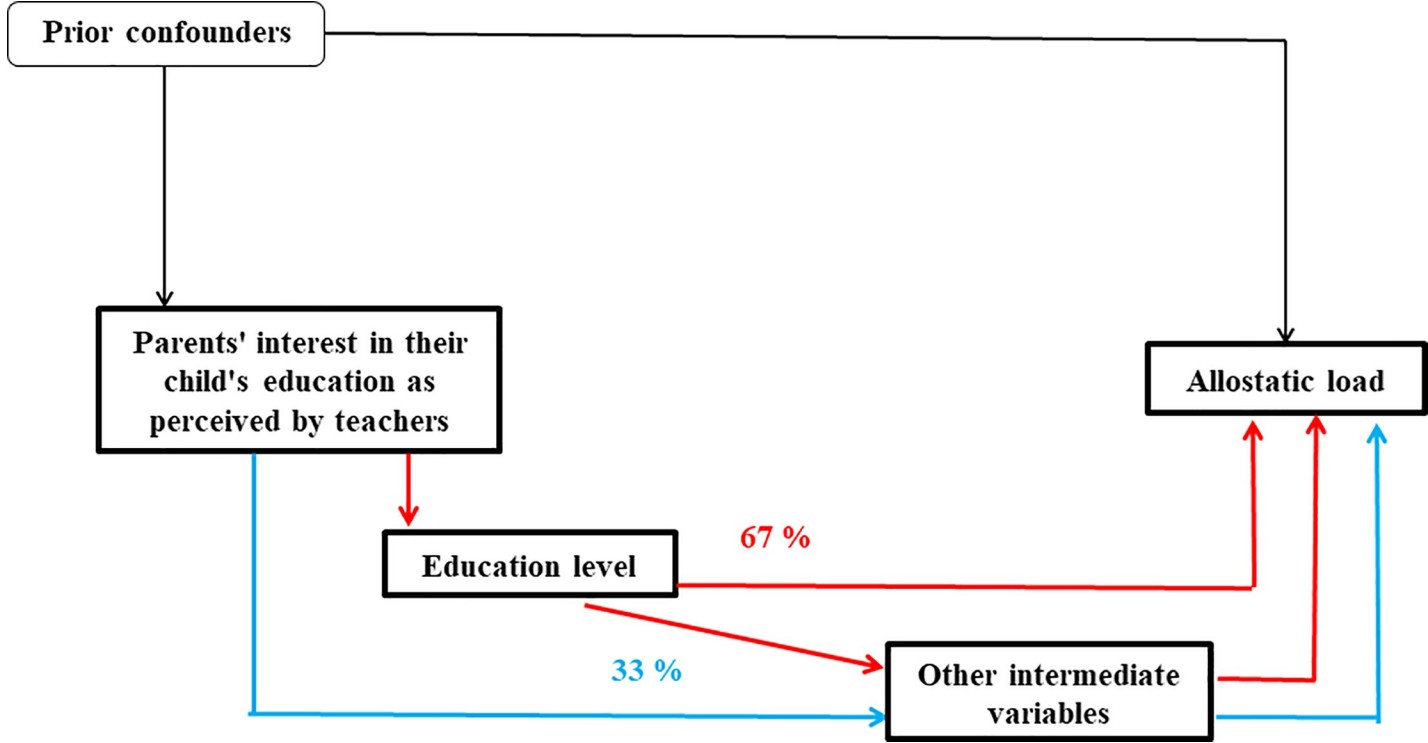

**Fig 2. Direct and indirect effect results between parental interest and allostatic load obtained from multiple imputation for men (n = 4057).**

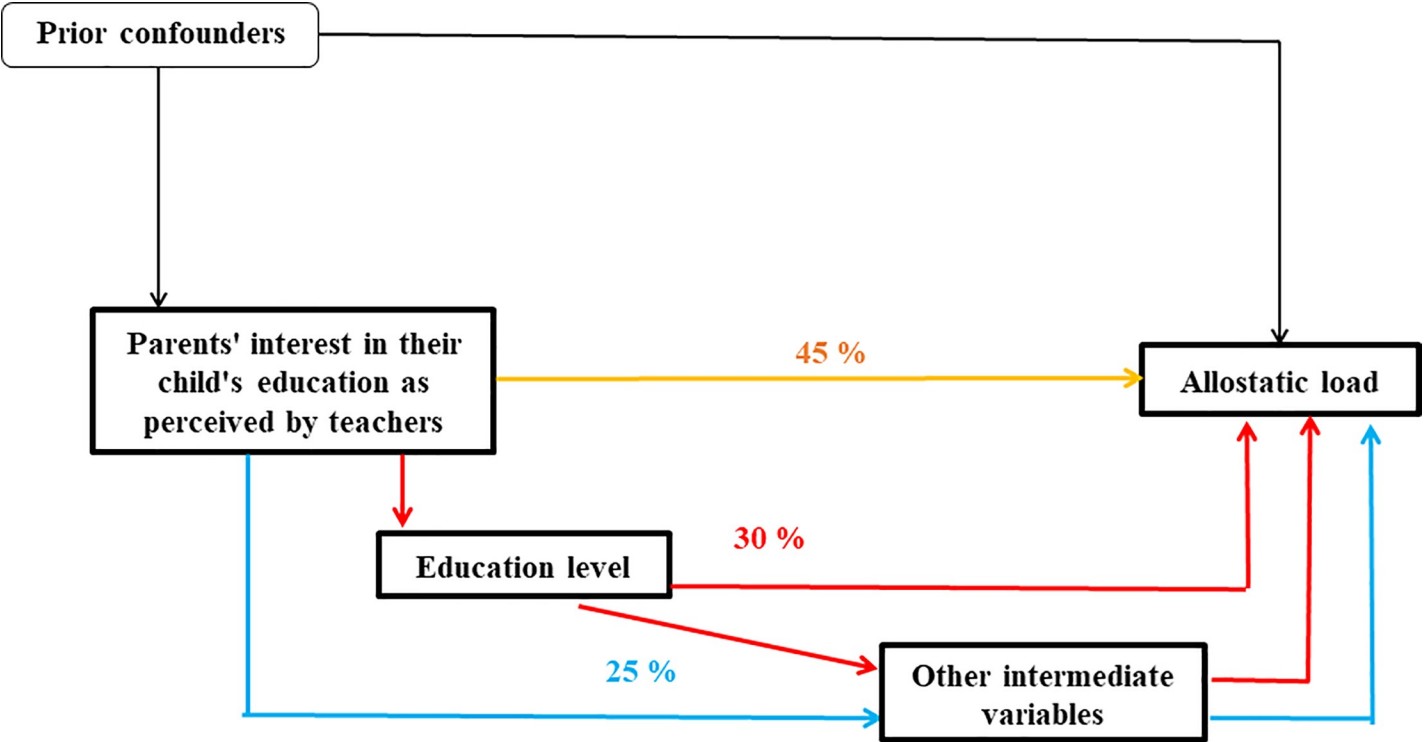

**Fig 3. Direct and indirect effect results between parental interest and allostatic load obtained from multiple imputation for women (n = 4 056).**

both men and women. Cohort members whose parents were perceived as uninterested or not very interested in their child's education, as reported by the children's teachers, had a higher allostatic load compared to individuals whose parents were considered to be interested. The association between parents' interest in their child's education as perceived by teachers and cohort member's physiological wear-and-tear operated through intermediate pathways over the life course. Among men, 67% of the association operated through the educational pathway, and 33% through the other variables including income, smoking, alcohol consumption and physical activity. Among women, only 30% of the association operated through the education pathway, 25% worked through the other variables in adulthood, including psychological distress. Much of the association (45%) was direct, and unexplained by the tested pathways. Our results are in line with other studies where parental interest in their offspring's studies as perceived by teachers was found to predict adult allostatic load and may buffer against poor mental health [30, 41]. Our findings provide insight into understanding how educational attainment as a reflection of dynamic life course social processes relates to physiological health, but also underline that parental-interest in children's education has not been given much attention in relation to health over the life course.

Our results may highlight an interplay between culture and biology [42, 43] whereby a tension between a child's home and school cultural environments may lead to a physiological stress response partly mediated by the educational trajectory. When a child attends school, the family social sphere meets the educational social sphere and if families have been socialized outside of the normative educational structure, they may need to adapt to the school environment [44]. Children who experience dissonance, as a chronically stressful challenge, may solicit their biological resources, experience multi-system physiological dysregulation as measured by allostatic load, and this embodiment may represent the cost of adaptation for the

children partly mediated by the educational trajectory. Our results are suggestive of this pathway, especially for boys/men.

We observed different associations in men and women. For men, the educational pathway had a significant and stronger effect on allostatic load, consistent with previously observed differences in mortality by educational level across age groups more pronounced in men than in women [3].

For women our results show that, after controlling for confounders and mediators, a sizable part of the initial effect remained unexplained. Based on our findings, we also hypothesize that dissonance affects the physiological health for girls/women directly, or through pathways that remain to be tested. The unexplained direct effect may represent other possible pathways, or different early life socialization and embodiment processes among girls. Intersecting domains of power including class, gender and others (race, disability, etc) are likely to be at play, and deserve further attention [45, 46].

Furthermore, it is possible that the behavior of teachers towards children whose parents they consider to be less involved, could be different. Teachers perceive families' economic and cultural capital once children enter school, and may unconsciously show favoritism toward students from more socially advantaged classes [47]. It is possible that our findings reflect a bias or difference whereby some teachers behaved differently towards children on this basis, contributing to increasing the stress of some children and therefore impacting their allostatic load.

An important aim of our analysis was to grasp the role of other intermediate factors through which parents' interest in their child's education as perceived by teachers may affect physiological processes. Adult income captured a large portion of the association for women and explained the association between educational level and allostatic load for men. Our findings suggest that consonant relationships between family and school, captured partly by parental interest, could promote ascending social mobility and therefore act as a vehicle towards social advantage, that may "buffer" the effects of an initially disadvantaged socio-economic environment on allostatic load [37, 48]. Furthermore, health behavior pathways appeared to explain a part of the association between parental interest and allostatic load for women and effects of education level on allostatic load for men. A consonant educational socialization could promote the embodiment of a health-relevant capital, i.e. the resources for acting in favor of health. Such consonance refers to all the "health related values, behavioral norm, knowledge and operational skills" [49]. However psychological malaise was found to explain the association between parental interest for women and allostatic load. Further analysis should be conducted in other cohorts to explore this association and ascertain its potential contextual specificity.

The main weakness of this study is that our variable measuring parental interest is one-sided, reflecting only the teacher's point of view. It would have been interesting to compare this measure with parents' perceptions; however, such data were unavailable. Attrition, and selection bias, common features related to longitudinal studies also pose issue. We carried out multiple imputation of missing data, a recommended method to avoid the interpretation of biased results, allowing them to be redressed to some extent. Information and recall biases may also be present, related to the self-reported nature of the data. With regard to alcohol consumption, we must consider that people with pathologies, but also those prone to alcohol addiction, are probably part of the abstention group, thus biasing the results of this group. Several years passed between the data collection sweeps wherein numerous life events most likely took place, which we cannot account for. Some variables in our study were measured at one given point in time, because we had only one measure available (i.e. allostatic load, sense of personal control, malaise), or we selected single variables as proxies for trajectories over time

(i.e. behavior, social position). It is a regret that there is no earlier measurement of allostatic load in order to analyze its dynamic changes over time. However, studies have shown that the kinetics of allostatic load as measured in adulthood remains generally constant over time [50]. Our choice of statistical models and included variables are based on *a priori* theoretical and conceptual considerations. We may have overlooked variables or assumptions contributing to the relationship between parental interest and allostatic load. Lastly, NCDS 58 is a UK cohort, with unique cultural and historical aspects. It is therefore necessary to take precautions when extrapolating our results.

Despite these limitations, this study has a number of strengths. It is a longitudinal population-based study containing prospectively collected data with great detail and breadth across the life span, allowing us to control for numerous confounding and mediating variables. Another important strength is in the sample size included in the biomedical survey, and the large number of biomarkers available.

Education is often used as a measure of social position, where higher educational attainment is associated with better health outcomes. Our findings suggest the importance of considering education as a product of early life interactions between family and school social spheres. The family cultural environment can be examined through the three dimensions of capital i.e. economic, social and cultural capital [51, 52]. Economic capital refers to the material resources and financial support, social capital concerns interpersonal support whereas cultural capital exists in three forms: embodied (e.g. values, skills), objectivized (e.g. cultural goods, books) and institutionalized (e.g. educational level). In the context of coexisting social spheres between family and the school environments, socio-cultural dissonance may occur. Indeed, "the standards of the school are not neutral; their requests for parental involvement may be laden with the cultural experiences of intellectual and economic elites" [53]. Among the socially disadvantaged, who may not necessarily possess a common language with which to negotiate with representatives of the school institution, educational success at school may indicate a conversion of their cultural capital, a phenomenon described as "acculturation". Conversely, for more socially advantaged students, this progression would be the result of the mobilization of their cultural capital heritage [54]. Consequently, the dissonance between the family social sphere and the school environment may lead to an "educational acculturation", requiring the family to assimilate to the new educational culture.

## 5. Conclusion

Parents' interest in their child's education as perceived by teachers measured during childhood was associated with physiological wear-and-tear in mid-life in both men and women. This may be due to a physiological stress response, induced from early life due to a possible dissonance between family and school cultural environments, which have lasting effects on health, through pathways including educational attainment, particularly in men. These results suggest that awareness of children's socio-cultural environments and gender should be considered when developing school or educational policies. As such, understanding family educational culture, cultural capital and socioeconomic position may contribute to developing adapted public policies supporting early childhood environments to reduce social inequalities in health.

## Supporting information

**S1 Table. Sensitivity analyses parental interest in 4 categories on complete-case data for men and women.**
(DOCX)

**S2 Table. Sensitivity analyses imputing parental interest measurement vs complete-case parental interest measurement for men and women.**
(DOCX)

**S3 Table. Descriptive characteristic on the subsample for men and women of the non-imputed data (n = 8 113).** Abbreviations and symbols: n = number of people; med = median; p25 = 25e percentile; p75 = 75e percentile. Values corresponding to the categories of allostatic load: Low: [0–2]; Medium: [3–4]; High: [5–12]. *P-values were calculated using a chi-squared test for categorical variables and Student t-test for the continuous variable.
(DOCX)

**S1 File. Detail on variable constructions.**
(DOCX)

**S2 File. Direct and indirect effect.**
(DOCX)

## Acknowledgments

We are grateful to the Centre for Longitudinal Studies (CLS), Institute of Education for the use of the NCDS data for making them available. We are grateful to the reviewers for their insightful comments and the time and effort that they devoted to this paper. Many thanks to Alexandra Soulier for her comments and thoughts on previous versions of this paper.

## Author Contributions

**Conceptualization:** Camille Joannès, Raphaële Castagné, Cyrille Delpierre, Michelle Kelly-Irving.

**Data curation:** Camille Joannès.

**Formal analysis:** Benoit Lepage.

**Methodology:** Benoit Lepage.

**Project administration:** Michelle Kelly-Irving.

**Supervision:** Michelle Kelly-Irving.

**Writing – original draft:** Camille Joannès, Raphaële Castagné, Benoit Lepage, Cyrille Delpierre, Michelle Kelly-Irving.

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
