## [Decision Letter · Decision Letter 0]

9 Oct 2020

PONE-D-20-18411

Could teacher-perceived parental interest be an important factor in understanding how education relates to later physiological health? A life course approach

PLOS ONE

Dear Dr. Camille Joannès,

Thank you for submitting your manuscript to PLOS ONE. After careful consideration, we feel that it has merit but does not fully meet PLOS ONE’s publication criteria as it currently stands. Therefore, we invite you to submit a revised version of the manuscript that addresses the points raised during the review process.

All comments from reviewer #1 must be addressed.

We look forward to receiving your revised manuscript.

Kind regards,

Mary Hamer Hodges, MBBS MRCP DSc

Academic Editor

PLOS ONE

Additional Editor Comments:

Please carefully revise this manuscript to enable the reader to better understand your hypothesis and how you can draw meaningful conclusions when the follow-up rate is so low.

Journal Requirements:

Reviewers' comments:

Reviewer's Responses to Questions

**Comments to the Author**

1. Is the manuscript technically sound, and do the data support the conclusions?

Reviewer #1: Yes

2. Has the statistical analysis been performed appropriately and rigorously? 

Reviewer #1: Yes

3. Have the authors made all data underlying the findings in their manuscript fully available?

Reviewer #1: Yes

4. Is the manuscript presented in an intelligible fashion and written in standard English?

Reviewer #1: No

5. Review Comments to the Author

Reviewer #1: I think this is an interesting manuscript but I feel that there is a need for the author to carefully analyze and present the paper more clearly to make it suitable for readers.

Some other comments are:

The paper is a little hard to read and the ideas don’t seem to flow sequentially and the quality of the language needs to be improved.

Some technical terms need to be adequately explained from the start: AL, PI ‘wear and tear’

The author failed to write on the relevance of the study, what is the gap there s/he wants to fill, not a sentence on assumptions about knowledge/epistemology of the paper, there was no literature in the introduction section triangulating with other studies and what is the related implication to programs. The author seems to be focusing more on the sociological aspect instead of articulating and linking the social activities of PI and AL to the child’s healthy or future outcomes.

6. PLOS authors have the option to publish the peer review history of their article (what does this mean?). If published, this will include your full peer review and any attached files.

Reviewer #1: No

---

## [Author Response · Author response to Decision Letter 0]

9 Dec 2020

Reviewer 1

1. I think this is an interesting manuscript but I feel that there is a need for the author to carefully analyse and present the paper more clearly to make it suitable for readers.

Author response: We thank the reviewer for this positive feedback on our work. We hope that the revised version of the manuscript will address the reviewer’s concerns on its clarity. More specifically, in the introduction section we have articulated the different conceptual stages from education to physiological wear and tear to health, considering the origins of educational attainment through the lens of interactions between families and school. We have also clarified the variable names and categories used.

Notably, p. 3 (lines 54-56), we have added a section to develop the inconsistency of the findings about the mechanisms through which education relates to health: “Beyond these groups of mechanisms others have been examined, such as environmental exposures and material conditions (5,6), calling for further studies to examine the mechanisms through which education relates to health“. In addition, p.4 (lines 82-87) we have explained the association of early life exposures and physiological wear-and-tear, mediated by education: “the social environment in early life may therefore have lasting effects on different social, biological, and behavioral factors that might act as mechanisms connecting education to repeated stress, which in turn affects health (20). Because educational attainment is the end point of a complex process where a wide array of factors (institutional, interpersonal, individual) shape trajectories of schooling, there is a need to look upstream in order to capture education as a long-term process grounded in a broader social context (20)”. We finally focused p.5 (lines 92-102) on parental interest in their child's education as an early life factor to examine “how the relationship between the home and school environments in early life affects physiological wear-and-tear, through educational attainment” (lines 100-102).

2. The paper is a little hard to read and the ideas don’t seem to flow sequentially and the quality of the language needs to be improved. Some technical terms need to be adequately explained from the start: AL, PI ‘wear and tear’

Author response: In the revised version of the manuscript, we have completely reviewed the language. We have also explained AL, PI and ‘wear and tear’ from the start. Regarding AL and wear and tear (p.3 lines 63-70) we have explained that these concepts refer to the embodiment process: “interactions (which) are perceived through the senses, are interpreted by the central nervous system leading to peripheral physiological responses. These physiological responses are adaptive processes, which maintain physiological stability in response to environmental challenges (10). The repeated activation of compensatory physiological mechanisms as a result of chronic exposure to stress can lead to physiological wear-and-tear, termed allostatic load (AL). AL measures the consequence of a prolonged activation of the stress response system by external challenges, leading to physiological imbalances across systems”. 

With regard to PI (parents' interest in their child's education as perceived by teachers), p.8 (lines 174-175) it “was measured at age 7, 11 and 16 using information provided by the child’s teachers. The teacher was asked to report the level of interest of each parent in their child’s education”. Also, p. 5 (lines 105-108) “Teacher’s assessments [...] may partly reflect the position and viewpoint of the educational institution in terms of children’s compliance with academic requirements and potentially capture the tension between the home and school environments experienced by some children”. 

3. The author failed to write on the relevance of the study, what is the gap there s/he wants to fill, not a sentence on assumptions about knowledge/epistemology of the paper, there was no literature in the introduction section triangulating with other studies and what is the related implication to programs. The author seems to be focusing more on the sociological aspect instead of articulating and linking the social activities of PI and AL to the child’s healthy or future outcomes.

Author response: Thank you for this constructive comment. In the revised version of the manuscript, we have amended the text to carefully highlight the gap our study aims to fill. More specifically, we have detailed by which mechanisms educational attainment is related to health (p.3 first paragraph) showing that part of the social gradient in health remains to be explained. We have added literature triangulating with other studies (p.5 lines 94-99): “parental interest in their child’s education has also been reported to have positive effects on psychosocial adjustment (22) later mental health (23) and metabolic outcomes in middle-age (24). Moreover, parental interest in their offspring’s schooling has been identified as a determinant of educational success (25–28). This parental involvement in children’s academic socialisation has been shown to influence academic success over and above child's social class and cognitive abilities (29).” 

We also added a section regarding the knowledge gap we want to fill (p.5 lines 114-122) “To our knowledge, only one previous study examined the association between parental academic involvement as perceived by teachers and AL in midlife in a Swedish cohort. Parental interest in their children’s studies during the last year of school, rather than parental social class or availability of practical academic support, was found to predict adult AL largely mediated by academic achievement, which the authors attributed to the potential consequence of physiological stress over the life course (31). Our study aims to examine the association between parents' interest in their child's education as perceived by teachers, during a child’s educational trajectory, and multi-system physiological dysregulation measured using allostatic load, and whether education and other pathways mediate this relationship.”

Reviewer 2

ABSTRACT

1. Line 27: Which child education are you referring? Is it the teacher's child or the pupils?

Author response: We agree with the reviewer that the formulation ‘teacher-perceived parental interest in their child’s education’ was rather confusing. In the revised version of the manuscript, it has been replaced by ‘parents' interest in their child's education as perceived by teachers’. 

2. Line 30: Who are these participants? You have to make it clear from the onset.

Author response: These participants are those who originally had AL scores available. We have modified the sentence to clarify this point p.2 (lines 30-31) “We used data from 9 377 women and men born during in 1958 in Great Britain and included in the National Child Development Study to conduct secondary data analyses” and in the section method p.7 (lines 135-137) “Our study is based on secondary analyses of data from the 1958 National Child Development Study (NCDS), an observational prospective population-based cohort study, which included all live births in Great Britain during one week in 1958 (n = 18 555)”.

3. Line 30: You are the data analysis, sample size and results in one sentence. i will suggest: Linear regression analyses for a 7850... Result revealed that people (be specific, who are these people?)...

Author response: We have amended the abstract on this sense p2 (lines 34-35). “Linear regression analyses were carried out on a sample of 8 113 participants with complete data for AL, missing data were imputed”. 

4. Line 35: why do you use conditional word 'may', your findings should be able to give answers to this hypothesis.

Author response: Thank you for this suggestion. We have modified this section to provide clear answers to our hypothesis. However, in the case of our study, it seems slightly out of scope to do not use terms that nuance because this study is a piece of evidence, with data limitations, and it would require further research to confirm the conceptual theories mobilized.

5. Line 36: line 35 and 36 are unclear, please rephrase. 

Author response: This comment is linked with the previous one. We have accordingly modified this section.

INTRODUCTION

6. Line 46: ...However, the mechanisms through which education relates to health, remain poorly understood...reference for this please.

Author response: Thank you, in line with this comment and reviewer 1’s, we have re-written the first paragraph of the introduction to give more details on the mechanisms by which education is associated with health and illustrating that this relationship is not entirely explained. In the literature, several mechanisms are highlighted by which education can affect health. However, when these variables are taken into account in different models, the association between education and health is affected but not fully explained, suggesting the existence of other pathways to be explored : “Gallo et al. showed that health behaviors and lifestyle factors explained only a part of the educational inequalities in total mortality (3). Similarly access to healthcare had a modest role in the educational gradient in health (4). Beyond these groups of mechanisms others have been examined, such as environmental exposures and material conditions (5,6), calling for further studies to examine the mechanisms through which education relates to health “(p.3 lines 51-56).

7. Line 60: How? please try to make your message reflective to the reader instead of being abstract. try to bring or collaboration in your thought... for eg: People with low socioeconomic conditions, i.e. as defined by low income, education and occupation, are at higher risk of infection compared to people with higher socioeconomic conditions.

Author response: We have added a sentence in the introduction (p.4 lines 77-79) in order to clarify this message “One recent study documented that disadvantaged early life socioeconomic conditions are associated with an increased risk of having a higher AL in midlife, mainly through educational pathways (17).”

8. Lines 61-65: sentence is too long. lots of repetition of 'wear and tear'...try to use other synonyms such as repeated stress or physiological consequence and more as appropriate. 

Author response: We reformulated lines 80-84 to have a clearer and less repetitive sentence “As such, educational attainment “is an excellent marker of the ‘healthfulness’ of accumulated childhood experience’’(19) as the social environment in early life may therefore have lasting effects on different social, biological, and behavioral factors that might act as mechanisms connecting education to repeated stress, which in turn affects health”.

9. from line 48-80 are confusing. The early life socioeconomic conditions and associations have not been articulated well. this link might help: https://equityhealthj.biomedcentral.com/articles/10.1186/s12939-017-0553-7

Author response: Thank you for this useful reference which has helped us restructure our presentation of the paper in the introduction & methods to clarify what we mean by early life socioeconomic conditions and associations (p.4). 

10. Line 88: the author fail to write on the relevant of the study, what are the gap there you want to fill, what is the related implication to policies and programs. 

Author response: We have now incorporated a section p.4 to develop the relevance of the study and the gap in knowledge we want to fill. We aim to contribute to the scientific literature about relevant pathways from the early life environment towards health outcomes in adulthood. This scientific evidence may inform future research about intervention programs and policies however our study is not adapted to be directly interpreted for policy purposes.

11. Line 95: the introduction does not have literature on important factor in understanding how education relates health. Does not triangulate with other literatures across the region. The literature in unclear on the factors affecting PI, education and heath outcomes. See: https://www.unaids.org/sites/default/files/sub_landing/files/10_4-Intro-to-triangulation-MEF_0.pdf

Author response: We thank the reviewer for this helpful document. In the revised version of the manuscript, we extended the first paragraph to better illustrate how education is related to health. Additionally, most of the introduction has been re-written in order to better justify the use of PI as an upstream determinant of educational attainment that may affect the allostatic load and in-turn impact health. 

METHOD

12. Line 110: The methods section does not have any structure. It is hard to identify the study design (desk review or quantitative or mix method?). What sampling strategy was employed in selecting participants for the study? 

Author response: We have restructured and clarified the methods section. The sampling strategy and participant selection that was used to constitute the NCDS cohort study is provided in more details on p.7, section ‘Study population’. 

13. Line 111: please state the criteria inclusion and exclusion

Author response: As detailed above, the study population section has been modified to give a detailed description of the participants included in the study. The flow chart (Figure 1) has been modified accordingly. 

14. Line 128: which data are we using here 'previous' or current. Where is the previous one?

Author response: Thank you for pointing out this confusion. The National Child Development Study is a national cohort study which was set up in the UK in 1958 and has been made accessible and open for analysis by scientists since its inception. There are therefore many previous analyses and papers based on the data from this cohort. To avoid any confusion, we’ve replaced ‘as used previously within this cohort’ by ‘based on tertiles in the population’. We have also clarified that we are conducting secondary data analysis on the cohort in the first sentence of the methods section (p.8 lines 167-169)

15. Line 132: Did you interview the parent of these children? If not, I think it would have been good to make them your Key informant thereby increasing the validity and reliability of your findings. see: https://methods.sagepub.com/book/key-concepts-in-social-research/n28.xml

Author response: As mentioned above, we have clarified that we are carrying out secondary data analysis on a UK national cohort study set up in 1958, therefore we were not in charge of data collection. We have also added the following sentence: ‘Data collection on health, economic, social and developmental factors was carried out on cohort members from birth until now at age 7, 11, 16, 23, 33, 42, 44/45, 46, 50, 55 and 62 years and conducted by the Centre for Longitudinal Studies.’ in the Study population section. 

We thank the reviewer to raise this point and agree that it would have been interesting to compare parents' interest in their child's education as perceived by teachers with a measure parental interest from the perspective of parents as key informants, this would have certainly increase the validity and reliability of our findings. Because no data on parental interest was collected from the parents themselves we could only focus on parental interest perceived by the teacher. This was mentioned as the main limitation of our study: “The main weakness of this study is that our variable measuring parental interest is one-sided, reflecting only the teacher’s point of view. It would have been interesting to compare this measure with parents' perceptions” (p.35 lines 450-452). 

16. Line 134: Were these scaled answers or open-ended answers. If open-ended questions were used, what was the process used to determine theses categories?

Author response: The teachers were asked the following closed question about each parent “With regards to the child’s educational progress, do the father and mother appear:” The teacher could then provide one of the following closed responses: “Over concerned about the child’s progress and/or expecting too high a standard ; Very interested ; To show some interest ; To show little or no interest ; Can’t say ; Inapplicable (e.g. no father)”. We have modified the sentence on this on p.8 (lines 174-177).

17. Line 156: these can be summed in a table as 'Participant category'

Author response: We reformulated the section using “Prior confounders” and added information regarding the corresponding variables “We selected variables from questionnaires completed during childhood by parents of cohort members that were likely confounders between PI and AL, based on the literature” (p.9 lines 191-192)

18. Line 161: was any demographic data collected? how many children studied?

Author response: The NCDS study information has been clarified p7. We have a sample of 9 377 participants who were followed up from birth, throughout childhood and into adulthood. The cohort study contains a vast number of variables on each individual across their life course. More information can be found here https://cls.ucl.ac.uk/cls-studies/1958-national-child-development-study/

19. Line 179: You must include a section on assumptions about knowledge / epistemology in your paper. The epistemological approach should be stated and referenced. How and why the approach applies to the study should be present, it should be made clear why this approach was the most appropriate for the study. There should be a brief discussion of how the research will use the approach.

Author response: We have, accordingly, modified this section (p.10-11, lines 211-240) to emphasize this point. “In order to determine whether any observed associations between PI and AL were due to subsequent adult intermediate factors, we selected four intermediate groups of life course variables as pathways between PI and health, based on epidemiological evidence and on empirical studies. The following mediating factors were then added to the models: i) Lynch et al. (37) suggested that health is the result of an accumulation of experiences and exposures due to the material world. PI could promote a positive accumulation of academic and social success (38), reducing the impact of economic difficulties on educational attainment […] ii) Tension between home and school environments experienced by some children may provide an insecure social environment, contributing to health through psychological processes. Indeed, the resulting negative emotions through psycho-neuroendocrine mechanisms could lead to a poor health status […] iii) The adoption of health behaviors is closely related to education with people with a higher education level are more likely to have protective health behaviors, impacting physiological functioning (17,41) […] We defined the sets of mediators according to the temporal and causal assumptions of the life course approach, applying the general approach of Baron & Kenny (42). We assumed the set health behaviors (at 42y) came after and could be influenced by socioeconomic position (at 33y), which came after and could be influenced by psychosocial condition (at 23y), which came after and could be influenced by education (at 23y), which came after and could be influenced by PI.”

20. Line 223: Please try to make your ideas flow succinctly from top to bottom...

Author response: We have modified the format of the presentation of the different stages of the statistical analysis and moved a theoretical paragraph upstream of the statistical analysis section (p.12). We hope that this allows a clearer reading.

RESULT

21. Line 236: can you describe the men and women in this study.

Author response: We have added an additional table (S5_Table C) in the Supplementary information that describes the characteristics of the men and women included in our study sample. In the revised version of the manuscript, a description of this new Supplementary Table has been added (p. 14 lines 290-296) 

22. Line 256: the results need to be discussed and interpreted with each of the confounders, making linkages or association and correlations with other variables. The author need to do comparison with PI vs AL and correlated with other factors.

Author response: We have made changes throughout the text of the results p.14-15 to detail our results according to the statistical associations between AL or PI and the other variables. 

23. Line 268: You should be able to explain and interpret your findings in this 

Author response: In this section, we focussed on describing the most salient results relating to our hypotheses. We felt that the explanatory section interpreting our findings would be more appropriate at the beginning of the discussion and the interpretive section throughout the discussion text. 

24. Line 419: It fails to provide guidance/suggestions about how national policy-makers and other concerned actors, including ministry of Basic education, social welfare, health personnel, nongovernmental organizations and international development agencies, should proceed.

Author response: Thank you for this suggestion. While we think our findings are a relevant contribution to the literature to understand how the early life environment contributes to health dynamics over the life course, we do not aim to directly inform guidance to policy-makers. Rather, this work may inform other types of research set-up to examine interventions which could, in turn, inform policy-making. 

In addition to the above comments, all spelling and grammatical errors pointed out by the reviewers have been corrected. We look forward to hearing from you in due time regarding our submission and to respond to any further questions and comments you may have.

Sincerely,

---

## [Decision Letter · Decision Letter 1]

7 Apr 2021

PONE-D-20-18411R1

Could teacher-perceived parental interest be an important factor in understanding how education relates to later physiological health? A life course approach

PLOS ONE

Dear Dr. Joannès,

Thank you for submitting your manuscript to PLOS ONE. After careful consideration, we feel that it has merit but does not fully meet PLOS ONE’s publication criteria as it currently stands. Therefore, we invite you to submit a revised version of the manuscript that addresses the points raised during the review process.

I have been assigned to help with this manuscript as the original editor is no longer available. One reviewer who had already reviewed the original submission has also commented on the resubmission, for which we are grateful. In contrast to the reviewer's opinion, I think the paper is well suited at PLOS ONE after all comments have been sufficiently addressed.

The paper deals with an important research question of how education and physiological health may be influenced by family characteristics such as parental interest in their child's education. The limitations, e.g. allostatic load is measured at only one point in time, parental interest is only assessed as perceived as teachers, have been appropriately commented on.

After careful reading of the paper, I have noticed a few additional issues that were not raised in the first round of revision, which I mention below, and which need to be addressed before moving forward with this manuscript.

We look forward to receiving your revised manuscript.

Kind regards,

Anja K Leist, Professor Dr.

Academic Editor

PLOS ONE

Journal Requirements:

Additional Editor Comments (if provided):

- Personally, I usually recommend to abstain from all unnecessary abbreviations as the reader needs to remember these throughout the manuscript. So I would suggest to increase readability to say, "parents' interest in their child's education as perceived by teachers, mentioned as parental interest in the following", and speak of parental interest instead of PI; and of allostatic load instead of AL.

- I was pleased to see VanderWeele and Vansteelandt's method applied to test the putative mediators. If not absolutely necessary, I would remove references to Baron and Kenny and instead define mediators etc. in line with VanderWeele and Vansteelandt.

- I suggest to add to the abstract that the direct link between parental interest and allostatic load was completely mediated in men, but only partially mediated in women.

- The main set of results for men and women need to be carefully checked. For example, I would use p<0.001 or a manually adjusted p level to indicate significance, as otherwise family wise error rate (alpha inflation) would be a problem. Equally importantly, on p. 36, line 339 ff., it seems by adjusting for confounders, the main association between parental interest and allostatic load is rendered insignificant in men (the confidence interval includes the null, the p's are above 0.05). If I am reading this correctly, the results need to be reformulated to reflect the fact that the main association was not significant anymore in the adjusted models.

Minor issues:

- Please also use . instead of , for the p values, confidence intervals (decimal points)

- p. 6 line 130 - there is an incomplete sentence (a few more missing periods throughout the manuscript, please check if ends of sentences are complete).

Reviewers' comments:

Reviewer's Responses to Questions

**Comments to the Author**

1. If the authors have adequately addressed your comments raised in a previous round of review and you feel that this manuscript is now acceptable for publication, you may indicate that here to bypass the “Comments to the Author” section, enter your conflict of interest statement in the “Confidential to Editor” section, and submit your "Accept" recommendation.

Reviewer #1: All comments have been addressed

2. Is the manuscript technically sound, and do the data support the conclusions?

Reviewer #1: Partly

3. Has the statistical analysis been performed appropriately and rigorously? 

Reviewer #1: Yes

4. Have the authors made all data underlying the findings in their manuscript fully available?

Reviewer #1: Yes

5. Is the manuscript presented in an intelligible fashion and written in standard English?

Reviewer #1: No

6. Review Comments to the Author

Reviewer #1: This has improved compared to the first submission; however, I still think the authors need to improve on the sequential flow of the concept. I will also recommend that this paper is submitted to International Journal of Research and Scientific Innovation in Social sciences (RSIS) or a mental health journal as this is not a public health paper. It is kind of related to early childhood development or education and its answers some correlated questions to its impact on earning potential, midlife health and behavioural status but I still find it hard to conceptualize and put the jigsaw concisely.

Line 49-50 please remove the “harsh material condition”

Line 51-52 this sentence is unclear, …inequalities in total mortality of Who?

The study design/methodology is not well-presented to explain the results. Interpretation of the results and the discussion are too wordy and are difficult to understand. It needs more careful proofreading.

7. PLOS authors have the option to publish the peer review history of their article (what does this mean?). If published, this will include your full peer review and any attached files.

Reviewer #1: No

---

## [Author Response · Author response to Decision Letter 1]

5 May 2021

Editor

1. Personally, I usually recommend to abstain from all unnecessary abbreviations as the reader needs to remember these throughout the manuscript. So I would suggest to increase readability to say, "parents' interest in their child's education as perceived by teachers, mentioned as parental interest in the following", and speak of parental interest instead of PI; and of allostatic load instead of AL.

Author response: Thank you for this suggestion, we have amended the abbreviations throughout the text of the manuscript, to increase the readability, by using “parental interest” instead of PI, “allostatic load” instead of AL, and we have also added “Sense of personal control” instead if SOC.

2. I was pleased to see VanderWeele and Vansteelandt's method applied to test the putative mediators. If not absolutely necessary, I would remove references to Baron and Kenny and instead define mediators etc. in line with VanderWeele and Vansteelandt.

Author response: We have amended the manuscript to remove the reference to Baron and Kenny, and altered the paragraph on mediators in line with VanderWeele and Vansteelandt’s recommendation on page 11 line 240: “We defined the sets of mediators according to the temporal and causal assumptions of the life course framework. As such, we made a pragmatic methodological assumption about the temporal ordering of variables in order to examine the mediating pathways whereby the exposure variable (parental interest at age 7, 11 and 16) preceded the respondent’s education, which preceded the other intermediate variables (psychological distress, socioeconomic position and health behaviors)”.

3. I suggest to add to the abstract that the direct link between parental interest and allostatic load was completely mediated in men, but only partially mediated in women.

Author response: Thank you for this we have amended the abstract page 2 line 40.

4. The main set of results for men and women need to be carefully checked. For example, I would use p<0.001 or a manually adjusted p level to indicate significance, as otherwise family wise error rate (alpha inflation) would be a problem. Equally importantly, on p. 36, line 339 ff., it seems by adjusting for confounders, the main association between parental interest and allostatic load is rendered insignificant in men (the confidence interval includes the null, the p's are above 0.05). If I am reading this correctly, the results need to be reformulated to reflect the fact that the main association was not significant anymore in the adjusted models.

Author response: Thank you for pointing this out, we agree that the section highlighted was ambiguous. We have modified the formulation throughout the results section for the multivariate results for men to better reflect the non-significant association. Page 24 lines 345-347, after adjustment for prior confounder, the association between parental interest and allostatic load remained significant (Model 2, β= 0.18 [0.03; 0.32]; p-value= 0.015). However, from the model 3, the association was no longer significant (Model 3, β= 0.06 [-0.09; 0.21]; p-value=0.44). Therefore, we have added line 349 “which was no longer significant” and line 353 “After sequentially controlling for all time-ordered life course, SEP and health behaviors, the association between parental interest and allostatic load was attenuated and not statistically significant (Model 6, β= -0.01 [-0.16; 0.14]).”

5. Please also use . instead of , for the p values, confidence intervals (decimal points)

Author response: Thank you for alerting us to this error. We have replaced commas with decimal points in the main text and supplementary files.

6. p. 6 line 130 - there is an incomplete sentence (a few more missing periods throughout the manuscript, please check if ends of sentences are complete).

Author response: Thank you for pointing this omission out. We have carefully proofread the text for other omissions and for fluidity of language. Page 6 line 131: “PI is associated with allostatic load, and (ii) explore four pathways through which PI may be differentially embodied during childhood, adolescence and early adulthood, leading to physiological wear-and-tear, as measured by allostatic load” and page 13 line 285 “See S3 File for more information on the different steps of the analysis of estimated direct and indirect effects.”

Reviewer

1. This has improved compared to the first submission; however, I still think the authors need to improve on the sequential flow of the concept. I will also recommend that this paper is submitted to International Journal of Research and Scientific Innovation in Social sciences (RSIS) or a mental health journal as this is not a public health paper. It is kind of related to early childhood development or education and its answers some correlated questions to its impact on earning potential, midlife health and behavioural status but I still find it hard to conceptualize and put the jigsaw concisely.

Line 49-50 please remove the “harsh material condition”

Line 51-52 this sentence is unclear, …inequalities in total mortality of Who?

The study design/methodology is not well-presented to explain the results. Interpretation of the results and the discussion are too wordy and are difficult to understand. It needs more careful proofreading.

Author response: Thank you for your helpful comments. We think that understanding how early life conditions in both the family and school environment may relate to later health are considerations of interest to a broad scientific and health audience, therefore we hope to pursue publication in Plos One.

Line 51-52 We replaced “harsh material condition” by disadvantaged material condition

Line 54-55 We added the additional point that it was total mortality of members of European cohorts.

We also have clarified some part of the methods section. Page 9 line 194 “To examine the relationship between parental interest and allostatic load, prior confounding variables potentially associated with both parental interest and allostatic load were added to the multivariable models. We selected variables most likely to be social or biological confounding factors from questionnaires completed during childhood by parents of cohort members, based on the literature.” Then the types of confounding factors were added such as “Socioeconomic confounders” line 199 or “Confounding factors related to family education” line 204. 

Parts of the discussion section have been reorganized to make it easier to understand.

We look forward to hearing from you in due course regarding our submission and to respond to any further questions and comments you may have.

Sincerely,

---

## [Editor Report · Decision Letter 2]

18 May 2021

Could teacher-perceived parental interest be an important factor in understanding how education relates to later physiological health? A life course approach

PONE-D-20-18411R2

Dear Dr. Joannès,

We’re pleased to inform you that your manuscript has been judged scientifically suitable for publication and will be formally accepted for publication once it meets all outstanding technical requirements.

Kind regards,

Anja K Leist, Professor Dr.

Academic Editor

PLOS ONE

Additional Editor Comments (optional):

The authors have appropriately addressed all conceptual comments. However, a few shortcomings need to be taken care of. Please check particularly point (3), the results need to be presented correctly before the manuscript can be accepted for publication.

(1)There are still a few language errors (e.g. in the discussion, "We observed two different scenarii for each gender respectively" instead of "We observed different associations in men and women").

(2) Allostatic load is abbreviated as AL in at least two instances in the manuscript.

(3) The results section still presents insignificant associations that are "reduced" or "attenuated" which doesn't make sense:

"Controlling for educational attainment reduced the

349 strength of the association between parental interest and allostatic load which was no longer

350 significant (Model 3, β= 0.06 [-0.09; 0.21]). The association was only marginally affected when

351 psychological status at age 23 were accounted for (Model 4, β= 0.05 [-0.10; 0.21]).

352 Occupational social class and wealth reduced the strength of the association further (Model 5,

353 β= 0.02 [-0.13; 0.17]) with wealth making a significant contribution. After sequentially

354 controlling for all time-ordered life course, SEP and health behaviors, the association between

355 parental interest and allostatic load was attenuated and not statistically significant (Model 6, β=

356 -0.01 [-0.16; 0.14])."

I suggest to rewrite "The associations between parental interest and allostatic load were rendered insignificant after including educational attainment in Model 3 (Model 3, β= 0.06 [-0.09; 0.21]). Further adjusting for psychological status at age 23 (Model 4) and occupational social class and wealth (Model5) did not change the result patterns". I would not comment on the significant contribution of wealth in line with the Table 2 fallacy.
---

## [Editor Report · Acceptance letter]

9 Jun 2021

PONE-D-20-18411R2 

Could teacher-perceived parental interest be an important factor in understanding how education relates to later physiological health? A life course approach 

Dear Dr. Joannès:

I'm pleased to inform you that your manuscript has been deemed suitable for publication in PLOS ONE. Congratulations! Your manuscript is now with our production department. 

Kind regards, 

on behalf of

Prof. Dr. Anja K Leist 

Academic Editor

PLOS ONE